# GEOMETRIC-MEAN POLICY OPTIMIZATION

**Yuzhong Zhao**[1][*]  **Yue Liu**[1][*]  **Junpeng Liu**[2]  **Jingye Chen**[3]  **Xun Wu**[4]  **Yaru Hao**[4]
**Tengchao Lv**[4]  **Shaohan Huang**[4]  **Lei Cui**[4]  **Qixiang Ye**[1]  **Fang Wan**[1][†]  **Furu Wei**[4][†]
[1]UCAS   [2]CUHK   [3]HKUST   [4]Microsoft Research
https://aka.ms/GeneralAI

## ABSTRACT

Group Relative Policy Optimization (GRPO) has significantly enhanced the reasoning capability of large language models by optimizing the arithmetic mean of token-level rewards. Unfortunately, GRPO is observed to suffer from unstable policy updates when facing tokens with outlier importance-weighted rewards, which manifest as extreme importance sampling ratios during training. In this study, we propose *Geometric-Mean Policy Optimization* (**GMPO**), with the aim to improve the stability of GRPO through suppressing token reward outliers. GMPO is plug-and-play—simply replacing GRPO's arithmetic mean with the geometric mean of token-level rewards, as the latter is inherently less sensitive to outliers. GMPO is theoretically plausible—analysis reveals that both GMPO and GRPO are weighted forms of the policy gradient while the former enjoys more stable weights, which consequently benefits policy optimization and performance. Experiments on multiple mathematical reasoning benchmarks show that GMPO-7B improves the average Pass@1 of GRPO by up to 4.1%, outperforming many state-of-the-art approaches. Code is available at https://github.com/callsys/GMPO and verl.

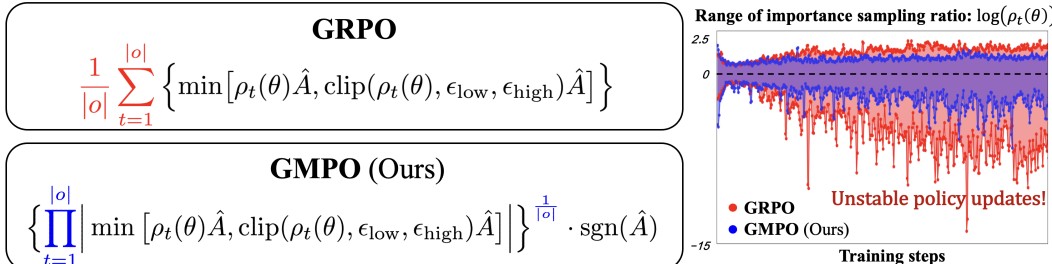

Figure 1: Comparison between GRPO and our GMPO. GRPO optimizes the arithmetic mean of token-level rewards while GMPO optimizes the geometric mean (left). When training with GRPO, the importance sampling ratio $\left(\rho_t(\theta) = \frac{\pi_\theta(o_t|q,o_{<t})}{\pi_{\theta_{\text{old}}}(o_t|q,o_{<t})}\right)$ frequently reaches extreme values, leading to unstable policy updates. In contrast, GMPO enjoys more stable importance sampling ratio and fewer outliers (right).

## 1 INTRODUCTION

As test-time scaling becomes a key research focus in the large language model community, recent post-training methods have increasingly sought to extend chain-of-thought (CoT) generation to enhance reasoning capabilities. Recent advances, such as Group Relative Policy Optimization (GRPO) (Shao et al., 2024), leverage multiple sampled responses per input prompt to compute relative rewards and advantages ($\hat{A}$ in Figure 1, left), leading to notable improvements in reasoning performance. By maximizing the arithmetic mean of token-level rewards, GRPO has achieved strong results on complex tasks such as mathematics, code generation, and multimodal reasoning.

---

[*] Equal contribution. [†] Corresponding authors. Work done during internship at Microsoft Research.

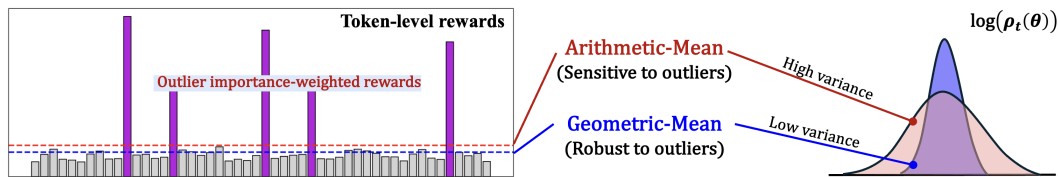

Figure 2: Compared to the arithmetic mean, the geometric mean is more robust to outliers and yields importance sampling ratio distributions with lower variance.

During GRPO training, the importance-weighted reward for each token is given by $\rho_t(\theta)\hat{A}$, where the importance sampling ratio $\rho_t(\theta)$ is defined as $\rho_t(\theta) = \frac{\pi_\theta(o_t|q,o_{<t})}{\pi_{\theta_{\text{old}}}(o_t|q,o_{<t})}$. This ratio plays a key role in PPO (Schulman et al., 2017) and GRPO, ensuring that policy updates are grounded in data from the current policy $\pi_\theta$. Large deviations of $\rho_t(\theta)$ from 1 indicate excessive policy shifts, leading to overly aggressive updates and instability. Constraining the ratio within a reasonable range is therefore critical for stable and reliable training.

As shown in Figure 1 (top left), the objective of GRPO involves the arithmetic mean of token-level rewards, which is sensitive to outliers (Figure 2). As training progresses (Figure 1, right), the range of $\rho_t(\theta)$ under GRPO expands, leading to unstable policy updates and degraded model performance. To mitigate this, GRPO applies a clipping range $(\epsilon_{\text{low}}, \epsilon_{\text{high}})$ to restrict large deviations of $\rho_t(\theta)$. However, this constraint leads to limited exploration and an early deterministic policy, which can hinder the scaling process (Yu et al., 2025).

To alleviate the instability while enhancing GRPO's exploration capability, we propose *Geometric-Mean Policy Optimization* (**GMPO**), as shown in Figure 1 (bottom left). GMPO takes full advantage of the geometric mean, which is inherently less sensitive to outliers and yields importance sampling ratio distributions with lower variance (Figure 2). During training (Figure 1, right), the range of GMPO 's $\rho_t(\theta)$ remains stable, exhibiting fewer extreme values than GRPO. With GMPO, we can maintain stable policy optimization while allowing a larger clipping range to promote greater exploration.

To further emphasize the advantages of GMPO, we provide detailed theoretical and experimental analyses to justify its training objective. First, we show that GMPO 's objective produces a narrower value range than GRPO's, indicating reduced training variance and more stable policy updates. Second, from a gradient perspective, GMPO provides a more balanced update signal and exhibits greater robustness to outlier values of the importance sampling ratio $\rho_t(\theta)$. Third, as training progresses, GMPO maintains a smaller KL divergence from the pre-trained model and higher token entropy than GRPO, indicating enhanced stability (via smaller KL) and greater policy exploration (via higher entropy).

Extensive experiments on language, multimodal, and agentic reasoning tasks demonstrate the advantages of GMPO over GRPO. Specifically, on five mathematical reasoning benchmarks of varying difficulty (AIME24 (Li et al., 2024), AMC (Li et al., 2024), MATH500 (Hendrycks et al., 2021), Minerva (Lewkowycz et al., 2022), and OlympiadBench (He et al., 2024)), GMPO improves the average Pass@1 accuracy by 4.1% (63.4% vs. 59.3%) using the Qwen2.5-7B model, compared to GRPO. In addition, GMPO improves the Pass@1 accuracy by 2.1% (96.7% vs. 94.6%) on MATH500 using the Qwen-32B (Yang et al., 2025) Mixture-of-Experts model. On the Geometry3K multimodal reasoning benchmark (Lu et al., 2021), GMPO increases the average Pass@1 accuracy by 1.4% (54.7% vs. 53.3%) with the Qwen2.5-VL-7B model. On the ALFWorld agentic reasoning benchmark (Shridhar et al., 2020), GMPO increases the overall accuracy by 13.1% (85.9% vs. 72.8%) with the Qwen2.5-1.5B model.

The contributions of this study are summarized as follows:

- We propose *Geometric-Mean Policy Optimization* (**GMPO**), which stabilizes the GRPO algorithm by maximizing the geometric mean of token-level rewards.

- We conduct thorough theoretical and empirical analyses, showing that GMPO improves stability while enhancing exploration relative to GRPO.

- GMPO-7B consistently outperforms GRPO-7B across diverse reasoning scenarios, delivering notable improvements in accuracy: 4.1% higher on five mathematical reasoning benchmarks, 1.4% higher on the Geometry3K multimodal reasoning benchmark, and a remarkable 13.1% improvement on the ALFWorld agentic reasoning benchmark.

## 2 BACKGROUND

### 2.1 RELATED WORKS

Reinforcement learning (RL) has become a key approach for post-training large language models (LLMs), with verifiable rewards enabling significant reasoning improvements, as demonstrated by DeepSeek-R1 (Guo et al., 2025a). Building on Proximal Policy Optimization (PPO) (Schulman et al., 2017), numerous variants have been developed to enhance efficiency and performance.

GRPO (Shao et al., 2024; Guo et al., 2025a) eliminates the need for computationally expensive value models while maintaining strong results across mathematics, coding, and QA benchmarks. GPG (Chu et al., 2025) further simplifies optimization by eliminating surrogate losses, critics, and KL constraints. Several extensions address rollout selection or bias correction: SRPO (Zhang et al., 2025c) uses history resampling, DAPO (Yu et al., 2025) employs dynamic sampling, Dr.GRPO (Liu et al., 2025) mitigates length bias, and OPO (Hao et al., 2025) provides an optimal baseline to reduce gradient variance. Reward shaping and advantage estimation are also actively explored. EMPO (Zhang et al., 2025b) incorporates semantic entropy, AAPO (Xiong et al., 2025a) introduces advantage momentum, and BNPO (Xiao et al., 2025) adaptively normalizes rewards via a beta distribution. Seed-GRPO (Chen et al., 2025) scales policy updates by question uncertainty, while GRPO-lead (Zhang & Zuo, 2025) addresses reward sparsity through length-dependent accuracy, explicit penalties, and difficulty-aware reweighting. Efficiency-driven methods include CPPO (Lin et al., 2025) (pruning low-advantage completions), S-GRPO (Dai et al., 2025b) (early exit to cut redundancy), Ada-GRPO (Wu et al., 2025) (adaptive reasoning formats), and GVPO (Zhang et al., 2025a) (analytical KL-constrained weighting). GRPO-$\lambda$ (Dai et al., 2025a) dynamically switches between length-penalized and length-agnostic rewards to avoid collapse. Further methods improve rollout usage. PODS (Xu et al., 2025) trains only on informative subsets of parallel rollouts, while RePO (Li et al., 2025) retrieves diverse off-policy samples via replay. RAFT (Xiong et al., 2025b) trains solely on positive samples yet rivals GRPO. INTUITOR (Zhao et al., 2025) eliminates external rewards by using model self-certainty, and PRIME (Cui et al., 2025a) provides a scalable RL framework for reasoning. Exploration-focused techniques include the 80/20 rule (Wang et al., 2025), which emphasizes high-entropy minority tokens, and entropy-based advantage augmentation (Cheng et al., 2025). Complementary to algorithmic advances, data-centric approaches have also proven crucial. Open-Reasoner-Zero (Hu et al., 2025) curates 129k diverse, high-quality samples with curriculum learning. Eurus (Yuan et al., 2024) contributes a large-scale alignment dataset and novel reward modeling.

Despite rapid progress, the stability of RL for LLMs remains rarely explored, even though it is essential for developing reliable and scalable post-training systems. While several GRPO variants enhance stability through better baseline estimation (OPO (Hao et al., 2025)), reward shaping (GRPO-lead (Zhang & Zuo, 2025)) or reward normalization (BNPO (Xiao et al., 2025)), the underlying stability of the RL process remains a persistent challenge. Our work offers a complementary perspective on these methods by introducing a robust aggregation operator for token-level rewards, providing an orthogonal approach to achieving more reliable and scalable post-training systems.

### 2.2 PRELIMINARY

The Group Relative Policy Optimization algorithm was initially proposed in DeepSeek-math (Shao et al., 2024). The core idea is to estimate the baseline using relative rewards within a group of rollouts, which reduces the computational cost of the critic model and improves training stability. Specifically, for each question $q$ from the training set $Q$, GRPO samples a group of rollouts $\{o_1, o_2, \cdots, o_G\}$ from the old policy $\pi_{\theta_{\text{old}}}$ and calculates the corresponding rewards $\{r_1, r_2, \cdots, r_G\}$. Then the policy

model $\pi_\theta$ is optimized by maximizing the following objective:

$$\mathcal{J}_{\text{GRPO}}(\pi_\theta) = \mathbb{E}_{q \sim \mathcal{Q}, \{o_i\}_{i=1}^G \sim \pi_{\theta_{\text{old}}}(\cdot|q)}$$

$$\frac{1}{G} \sum_{i=1}^G \frac{1}{|o_i|} \sum_{t=1}^{|o_i|} \left\{ \min\left[\rho_{i,t}(\theta)\hat{A}_i, \text{clip}(\rho_{i,t}(\theta), \epsilon_{\text{low}}, \epsilon_{\text{high}})\hat{A}_i\right] - \beta \text{D}_{\text{KL}}(\pi_\theta \| \pi_{\text{ref}}) \right\}, \quad (1)$$

where $\rho_{i,t}(\theta) = \frac{\pi_\theta(o_{i,t}|q,o_{i,<t})}{\pi_{\theta_{\text{old}}}(o_{i,t}|q,o_{i,<t})}$ and $\hat{A}_i = \frac{r_i - \text{mean}(\{r_1, r_2, \cdots r_G\})}{\text{std}(\{r_1, r_2, \cdots r_G\})}$. $\rho_{i,t}(\theta)$ represents the importance sampling ratio of the $t$-th token in the $i$-th rollout based on the current policy $\pi_\theta$ and old policy $\pi_{\theta_{\text{old}}}$. $\hat{A}_i$ is the advantage of the $i$-th rollout and is calculated by normalizing the rewards within the same group according to GRPO. $(\epsilon_{\text{low}}, \epsilon_{\text{high}})$ denote the clipping thresholds and $\text{D}_{\text{KL}}(\pi_\theta \| \pi_{\text{ref}})$ is the KL regularization term. Following Dr. GRPO (Liu et al., 2025), we ignore $\text{D}_{\text{KL}}(\pi_\theta \| \pi_{\text{ref}})$ for simplicity and memory saving. The objective of GRPO is equivalent to the arithmetic mean of token-level rewards (we ignore the clipping range term for clarity), which can be formatted as:

$$\mathcal{J}_{\text{GRPO}}^*(\pi_\theta) = \mathbb{E}_{q \sim \mathcal{Q}, \{o_i\}_{i=1}^G \sim \pi_{\theta_{\text{old}}}(\cdot|q)} \left[ \frac{1}{G} \sum_{i=1}^G \frac{1}{|o_i|} \sum_{t=1}^{|o_i|} \rho_{i,t}(\theta)\hat{A}_i \right]. \quad (2)$$

In practice, the rollouts are sampled from the old policy $\pi_{\theta_{\text{old}}}$. To approximate policy updates as if they were based on rollouts sampled from the current policy $\pi_\theta$, the normalized advantage $\hat{A}_i$ of each rollout is weighted by the importance sampling ratio $\rho_{i,t}(\theta)$.

## 3 GEOMETRIC-MEAN POLICY OPTIMIZATION

As shown in Figure 1(right), we observe tokens with extreme importance sampling ratios during GRPO training, indicating unreliable policy updates. This instability arises because GRPO's objective is sensitive to outlier values of importance-weighted rewards, which drive aggressive policy updates and further amplify the variance of importance sampling ratios.

To solve that, we propose *Geometric-Mean Policy Optimization* (**GMPO**), a stabilized variant of GRPO. Instead of optimizing the arithmetic mean of token-level rewards as shown in Equation 2, GMPO maximizes their geometric mean:

$$\mathcal{J}_{\text{GMPO}}^*(\pi_\theta) = \mathbb{E}_{q \sim \mathcal{Q}, \{o_i\}_{i=1}^G \sim \pi_{\theta_{\text{old}}}(\cdot|q)} \left[ \frac{1}{G} \sum_{i=1}^G \left( \prod_{t=1}^{|o_i|} |\rho_{i,t}(\theta)\hat{A}_i| \right)^{\frac{1}{|o_i|}} \cdot \text{sgn}(\hat{A}_i) \right], \quad (3)$$

where $\text{sgn}(\hat{A}_i)$ ensures the correct optimization direction, returning 1 when $\hat{A}_i > 0$ and -1 otherwise. $\mathcal{J}_{\text{GMPO}}^*(\pi_\theta)$ has a narrower value range than $\mathcal{J}_{\text{GRPO}}^*(\pi_\theta)$, which can be derived as follows:

$$|\mathcal{J}_{\text{GMPO}}^*(\pi_\theta)| = \mathbb{E}_{q \sim \mathcal{Q}, \{o_i\}_{i=1}^G \sim \pi_{\theta_{\text{old}}}(\cdot|q)} \left[ \frac{1}{G} \sum_{i=1}^G \left( \prod_{t=1}^{|o_i|} |\rho_{i,t}(\theta)\hat{A}_i| \right)^{\frac{1}{|o_i|}} \right]$$

$$\leq \mathbb{E}_{q \sim \mathcal{Q}, \{o_i\}_{i=1}^G \sim \pi_{\theta_{\text{old}}}(\cdot|q)} \left[ \frac{1}{G} \sum_{i=1}^G \frac{1}{|o_i|} \sum_{t=1}^{|o_i|} |\rho_{i,t}(\theta)\hat{A}_i| \right] = |\mathcal{J}_{\text{GRPO}}^*(\pi_\theta)|. \quad (4)$$

This narrower range suggests that the training process of GMPO experiences lower variance in the optimization objective, which can be viewed as evidence of more stable policy updates. Compared to $\mathcal{J}_{\text{GRPO}}(\pi_\theta)$, $\mathcal{J}_{\text{GMPO}}(\pi_\theta)$ is less sensitive to outliers because the geometric mean is inherently more robust to outliers than the arithmetic mean. As a result, $\mathcal{J}_{\text{GMPO}}(\pi_\theta)$ provides more reliable policy updates and maintains a more stable range of importance sampling ratios as shown in Figure 1(right). By expanding Equation 3 and incorporating the clipping range term from PPO (Schulman et al., 2017) at the token-level, we can derive the complete objective function of GMPO as follows:

$$\mathcal{J}_{\text{GMPO}}(\pi_\theta) = \mathbb{E}_{q \sim \mathcal{Q}, \{o_i\}_{i=1}^G \sim \pi_{\theta_{\text{old}}}(\cdot|q)}$$

$$\frac{1}{G} \sum_{i=1}^G \left\{ \prod_{t=1}^{|o_i|} \left| \min\left[\rho_{i,t}(\theta)\hat{A}_i, \text{clip}(\rho_{i,t}(\theta), \epsilon_{\text{low}}, \epsilon_{\text{high}})\hat{A}_i\right] \right| \right\}^{\frac{1}{|o_i|}} \cdot \text{sgn}(\hat{A}_i). \quad (5)$$

---

**Algorithm 1** GMPO Training objective

---

```python
def gmpo_loss(new_probs, old_probs, mask, advantage, epsilon=0.4):
    """
    new_probs [L, 1]: Token probabilities from the current model
    old_probs [L, 1]: Token probabilities from the old model
    mask [L, 1]: Indicates valid (non-padded) tokens
    advantage [1]: Advantage or normalized reward for the sequence
    epsilon [1]: Controls the clipping range
    """
    # Clipping at token-level & Clipping wider
    new_log_probs, old_log_probs = torch.log(new_probs), torch.log(old_probs)
    sgn = 1.0 if advantage > 0 else -1.0
    signed_log_ratio = sgn * (new_log_probs - old_log_probs)
    clipped = torch.clamp(signed_log_ratio, -epsilon, epsilon)
    min_log_ratio = torch.min(signed_log_ratio, clipped)
    min_log_ratio = sgn * min_log_ratio
    # Geometric-Mean Policy Optimization
    importance_sampling_ratio = torch.exp(min_log_ratio[mask].sum()/mask.sum())
    loss = -advantage * importance_sampling_ratio
    return loss
```

---

GMPO is straightforward to implement, and its pseudo-code is given in Algorithm 1. For numerical stability, both the product and clipping operations in Equation 5 are carried out in log space.

To better understand why GMPO is more stable than GRPO, we show that GMPO is more robust to tokens with extreme importance sampling ratios from a gradient perspective. Specifically, given question $q$ and rollout $o_i$, the gradients of $\mathcal{J}^*_{\text{GRPO}}(\pi_\theta)$ (Equation 2) and $\mathcal{J}^*_{\text{GMPO}}(\pi_\theta)$ (Equation 3) with respect to the model parameter $\theta$ are as follows[1]:

$$\nabla_\theta \mathcal{J}^*_{\text{GRPO}}(\pi_\theta)\Big|_{q,o_i} = \frac{1}{G \cdot |o_i|} \sum_{t=1}^{|o_i|} \rho_{i,t}(\theta) \cdot \hat{A}_i \cdot \nabla_\theta \log(\pi_\theta(o_{i,t}|q, o_{i,<t})), \tag{6}$$

$$\nabla_\theta \mathcal{J}^*_{\text{GMPO}}(\pi_\theta)\Big|_{q,o_i} = \frac{1}{G \cdot |o_i|} \sum_{t=1}^{|o_i|} \Big( \prod_{k=1}^{|o_i|} \rho_{i,k}(\theta) \Big)^{\frac{1}{|o_i|}} \cdot \hat{A}_i \cdot \nabla_\theta \log(\pi_\theta(o_{i,t}|q, o_{i,<t})), \tag{7}$$

The term $\hat{A}_i \cdot \nabla_\theta \log(\pi_\theta(o_{i,t}|q, o_{i,<t}))$ quantifies the influence of the generated token $o_{i,t}$ on the parameters $\theta$, which corresponds to the standard policy gradient (Sutton et al., 1999). The gradients of both objectives are weighted sums of the policy gradients of the generated tokens, but with different weights. For $\mathcal{J}^*_{\text{GRPO}}(\pi_\theta)$, the weight of the token $o_{i,t}$ includes its individual importance sampling ratio $\rho_{i,t}(\theta)$. An extreme $\rho_{i,t}(\theta)$ will cause the token gradient to be too large or small, resulting in aggressive policy updates. For $\mathcal{J}^*_{\text{GMPO}}(\pi_\theta)$, the weight of the token $o_{i,t}$ includes the geometric mean of all the ratios $\Big( \prod_{k=1}^{|o_i|} \rho_{i,k}(\theta) \Big)^{\frac{1}{|o_i|}}$ in the same sequence, which provides a more balanced update signal and is more robust to outlier values.

Beyond the proposed training objective, we demonstrate the effectiveness of the following key designs in GMPO:

**(i) Clipping at token-level.** Unlike vanilla GRPO in DeepSeek-math (Shao et al., 2024), DeepSeek-R1 (Guo et al., 2025a) maximizes the sequence-level reward $(\prod_{t=1}^{|o_i|} \rho_{i,t}(\theta))\hat{A}_i$ and clips outliers at the sequence-level, i.e., $\text{clip}\big( \prod_{t=1}^{|o_i|} \rho_{i,t}(\theta), \epsilon_{\text{low}}, \epsilon_{\text{high}} \big)$. The term $\prod_{t=1}^{|o_i|} \rho_{i,t}(\theta)$ also appears in the objective of GMPO (Equation 3 and 5). However, instead of applying clipping at the sequence-level as in DeepSeek-R1, we find it more effective to perform clipping at the token-level, as shown in Equation 5. The rationale is as follows: (1) Clipping at the token-level is more stable than at the sequence-level. As shown in Figure 3, the sequence-level clip (GMPO-seq-clip-$(e^{-0.4}, e^{0.4})$) has a larger importance sampling range than the token-level clip (GMPO $(e^{-0.4}, e^{0.4})$), which makes it more prone to creating extreme gradients during optimization. (2) Sequence-level clipping is too aggressive compared to token-level clipping. Once triggered, it sets the gradients of all tokens in the sequence to zero, potentially discarding valuable update signals from informative parts of rollouts.

---

[1]The clipping range term is omitted for clarity. Detailed derivations are provided in Appendix A.

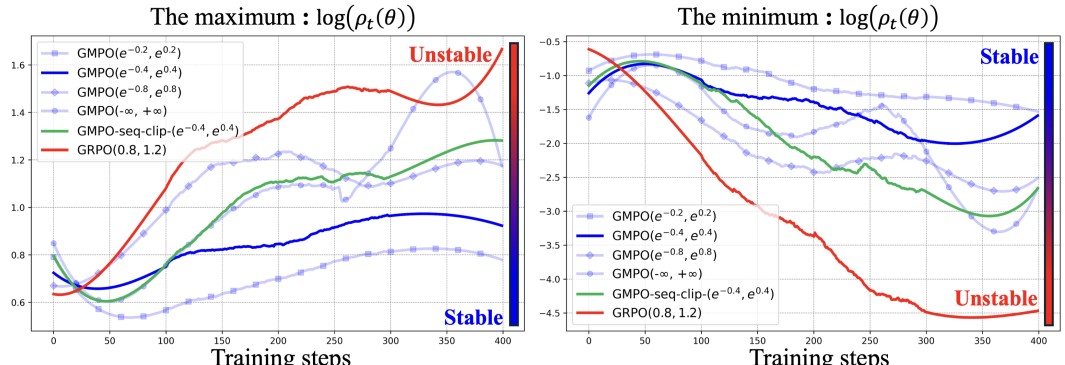

Figure 3: The range of importance sampling ratio $\rho_t(\theta)$ with respect to different clipping ranges and training steps. A wider range indicates less stable policy updates. Compared to GRPO with a clipping range of $(0.8, 1.2)$, GMPO demonstrates greater stability, even with a larger clipping range of $(e^{-0.4}, e^{0.4})$. All curves are smoothed for clarity.

**(ii) Clipping wider.** As illustrated in DAPO (Yu et al., 2025), the clipping operation can limit exploration and cause an early deterministic policy, which hinders the scaling process. To encourage exploration without sacrificing stability, DAPO uses a clip-higher strategy, which slightly expands the clipping range $(\epsilon_{\text{low}}, \epsilon_{\text{high}})$ from $(0.8, 1.2)$ to $(0.8, 1.28)$. As shown in Figure 1, we visualize the maximum and minimum importance sampling ratios at each training step for both GRPO and GMPO. The key observations are: (1) As training proceeds, the importance sampling ratios span a wider range, indicating more aggressive policy updates and increased instability. (2) Compared to GRPO, GMPO preserves a narrower range of importance sampling ratios, suggesting more stable updates. (3) For GMPO, expanding the clipping range from $(e^{-0.2}, e^{0.2})$ to $(-\infty, +\infty)$ increases instability in policy updates. Based on these findings, we balance training stability with exploration by setting clipping thresholds $(\epsilon_{\text{low}}, \epsilon_{\text{high}})$ in Equation 5 to $(e^{-0.4}, e^{0.4})$. This range is significantly larger than both GRPO and DAPO, encouraging greater exploration and improving performance.

## 4 EXPERIMENT

### 4.1 IMPLEMENTATION DETAIL

**Model.** We evaluate the algorithm's performance on both language-only, agentic, and multimodal reasoning tasks. For language-only tasks, following Dr.GRPO (Liu et al., 2025), we use Qwen2.5-Math-1.5B (Yang et al., 2024), Qwen2.5-Math-7B, DeepSeek-R1-Distill-Qwen-7B (Guo et al., 2025b) and Qwen3-32B (Yang et al., 2025) as our base models to assess performance on mathematical tasks. For agentic tasks, we use Qwen2.5-Instruct-1.5B (Yang et al., 2024) as the base model following GiGPO (Feng et al., 2025). For multimodal tasks, we use Qwen2.5-VL-Instruct-7B (Bai et al., 2025) as the base model to train GRPO and GMPO, and evaluate their performance on geometry reasoning tasks.

**Training.** For language-only tasks, following the setup of Dr.GRPO (Liu et al., 2025), we use MATH (Hendrycks et al., 2021) Levels 3–5 as the training dataset for models under 7B, which contains 8,523 mathematical problems. For each question, we generate 8 rollouts and cap the model's maximum response length at 3,000 tokens. During each RL training round, the old policy $\pi_{\theta_{\text{old}}}$ produces 1,024 rollouts, and the current policy $\pi_\theta$ is updated 8 times with a batch size of 128. For Mixture-of-Experts models, we use DeepScaleR (Luo et al., 2025) and CountDown (Pan, 2024) as the training dataset, with further details provided in Appendix E. For multimodal tasks, we follow the setup of EasyR1 (Zheng et al., 2025) and use Geometry3K (Lu et al., 2021) as the training dataset. For agentic tasks, we follow the setup of GiGPO (Feng et al., 2025) for training and inference, with further details provided in Appendix D. For mathematical problems, rewards are verifiable: "1" for correct responses and "0" for incorrect ones. Our method is mainly compared with Dr.GRPO and GRPO, under the same experimental setup as in Tables 1, and 2.

Table 1: Comparison of GRPO and GMPO for language models (a), multimodal models (b), Mixture-of-Experts models (c), and agentic models (d).

| Language Model | AIME24 | AMC | MATH500 | Minerva | Oly. | **Avg.** |
|---|---|---|---|---|---|---|
| GRPO-1.5B Shao et al. (2024) | 23.3 | 49.4 | 75.2 | 25.7 | 39.0 | 42.5 |
| GMPO-1.5B (Ours) | 20.0 | 53.0 | 77.6 | 30.1 | 38.7 | **43.9** |
| GRPO-7B Shao et al. (2024) | 40.0 | 59.0 | 83.4 | 32.4 | 41.3 | 51.2 |
| GMPO-7B (Ours) | 43.3 | 61.4 | 82.0 | 33.5 | 43.6 | **52.7** |
| GRPO-7B Shao et al. (2024) (R1-Distill) | 43.3 | 67.5 | 89.0 | 39.7 | 56.7 | 59.3 |
| GMPO-7B (R1-Distill, Ours) | 46.6 | 78.3 | 91.4 | 37.9 | 62.5 | **63.4** |

(a) Five mathematical reasoning benchmark.

| Multimodal Model | Geometry3K |
|---|---|
| GRPO-7B (Shao et al., 2024) | 53.3 |
| GMPO-7B (Ours) | **54.7** |

(b) Geometry3K benchmark.

| MoE Model | MATH500 |
|---|---|
| GRPO-32B (Shao et al., 2024) | 94.6 |
| GMPO-32B (Ours) | **96.7** |

(c) MATH500 benchmark.

| Agentic Model | Pick | Look | Clean | Heat | Cool | Pick2 | **ALL** |
|---|---|---|---|---|---|---|---|
| GRPO-1.5B Shao et al. (2024) | 85.3 | 53.7 | 84.5 | 78.2 | 59.7 | 53.5 | 72.8 |
| GMPO-1.5B (Ours) | 93.1 | 78.6 | 81.0 | 88.2 | 82.1 | 89.5 | **85.9** |

(d) ALFWorld benchmark.

**Evaluation.** We evaluate our method on five mathematical reasoning benchmarks of varying difficulty following Dr.GRPO (Liu et al., 2025), one multimodal reasoning benchmark following EasyR1 (Zheng et al., 2025), and one agentic reasoning benchmark. The mathematical benchmarks include: AIME24, which consists of 30 high-school-level olympiad problems from the American Invitational Mathematics Examination 2024; AMC, which contains 83 intermediate-difficulty multiple-choice problems; MATH500, a subset of 500 problems from the original MATH dataset covering algebra, geometry, and number theory; Minerva (Lewkowycz et al., 2022), which features 272 graduate-level problems requiring multi-step reasoning; and OlympiadBench (Oly.) (He et al., 2024), a collection of 675 high-difficulty olympiad problems. These benchmarks collectively cover a broad spectrum of problem types and difficulty levels. For multimodal reasoning, we use Geometry3K (Lu et al., 2021), a visual question answering dataset consisting of 601 geometry-focused problem-solving questions. For agentic reasoning, we evaluate on ALFWorld (Shridhar et al., 2020), an embodied environment designed to assess the ability of LLM agents to perform multi-step decision-making. We primarily use the Pass@1 metric for comparative analysis, which evaluates whether a single generated response to a problem meets the required criteria. For language tasks, we set the temperature to 0.0 and generate one answer per question, following Dr.GRPO (Liu et al., 2025). For the multimodal task, we set the temperature to 0.5 and generate 16 answers per question.

## 4.2 PERFORMANCE

Table 1 2 present a comprehensive evaluation of our GMPO approach against established reasoning methods across multiple benchmarks. Our method demonstrates consistent and substantial improvements over strong baseline systems.

**Language-only tasks.** GMPO demonstrates consistent improvements across different base models. With Qwen2.5-Math-1.5B, it achieves 43.9% average performance, outperforming GRPO by 1.4% and Dr.GRPO by 1.8%. Similar gains are observed with Qwen2.5-Math-7B (+1.5% vs. GRPO, +1.3% vs. Dr.GRPO) and DeepSeek-R1-Distill-Qwen-7B (+4.1% vs. GRPO, +1.9% vs. Dr.GRPO). In the stability-sensitive Mixture-of-Experts (MoE) setting with Qwen3-32B, GMPO achieves 96.7% accuracy on MATH500, outperforming GRPO by 2.1%. Additional results for MoE models are provided in Appendix E.

**Multimodal tasks.** Using Qwen2.5-VL-Instruct-7B as the base model, GMPO surpasses GRPO by 1.4% on Geometry3K, highlighting its potential for broader application in multimodal tasks.

Table 2: Comparison of GMPO and state-of-the-art methods on mathematical reasoning benchmarks.

| Model | AIME24 | AMC | MATH500 | Minerva | Oly. | Avg. |
|---|---|---|---|---|---|---|
| Qwen2.5-Math-1.5B (Qwen et al., 2025) | 16.7 | 43.4 | 61.8 | 15.1 | 28.4 | 33.1 |
| Qwen2.5-Math-1.5B-Instruct (Qwen et al., 2025) | 10.0 | 48.2 | 74.2 | 26.5 | 40.2 | 39.8 |
| Oat-Zero-1.5B (Liu et al., 2025) | 20.0 | 53.0 | 74.2 | 25.7 | 37.6 | 42.1 |
| GMPO-1.5B (Ours) | 20.0 | 53.0 | 77.6 | 30.1 | 38.7 | **43.9** |
| Qwen2.5-Math-7B (Qwen et al., 2025) | 16.7 | 38.6 | 50.6 | 9.9 | 16.6 | 26.5 |
| SimpleRL-Zero-7B (Zeng et al., 2025) | 26.7 | 60.2 | 78.2 | 27.6 | 40.3 | 46.6 |
| PRIME-Zero-7B (Cui et al., 2025a) | 16.7 | 62.7 | 83.8 | 36.0 | 40.9 | 48.0 |
| OpenReasoner-Zero-7B @ 3k (Hu et al., 2025) | 13.3 | 47.0 | 79.2 | 31.6 | 44.0 | 43.0 |
| OpenReasoner-Zero-7B @ 8k (Hu et al., 2025) | 13.3 | 54.2 | 82.4 | 31.6 | 47.9 | 45.9 |
| Eurus-7B (Yuan et al., 2024) | 16.7 | 62.7 | 83.8 | 36.0 | 40.9 | 48.0 |
| GPG-7B (Chu et al., 2025) | 33.3 | 65.0 | 80.0 | 34.2 | 42.4 | 51.0 |
| Oat-Zero-7B (Liu et al., 2025) | 43.3 | 62.7 | 80.0 | 30.1 | 41.0 | 51.4 |
| GMPO-7B (Ours) | 43.3 | 61.4 | 82.0 | 33.5 | 43.6 | **52.7** |
| Oat-Zero-7B (Liu et al., 2025) (R1-Distill) | 50.0 | 74.7 | 89.6 | 37.5 | 55.7 | 61.5 |
| GMPO-7B (R1-Distill, Ours) | 46.6 | 78.3 | 91.4 | 37.9 | 62.5 | **63.4** |

Table 3: Comparison of objectives and their performance under same training settings.

$$1 : \frac{1}{|o|} \sum_{t=1}^{|o|} \left( \min \left[ \rho_t(\theta)\hat{A}, \mathrm{clip}(\rho_t(\theta), \epsilon_{\mathrm{low}}, \epsilon_{\mathrm{high}})\hat{A} \right] \right)$$

$$2 : \left\{ \prod_{t=1}^{|o|} |\rho_t(\theta)\hat{A}| \right\}^{\frac{1}{|o|}} \cdot \mathrm{sgn}(\hat{A})$$

$$3 : \left\{ \left| \min \left[ (\prod_{t=1}^{|o|} \rho_t(\theta))\hat{A}, \mathrm{clip}(\prod_{t=1}^{|o|} \rho_t(\theta), \epsilon_{\mathrm{low}}, \epsilon_{\mathrm{high}})\hat{A} \right] \right| \right\}^{\frac{1}{|o|}} \cdot \mathrm{sgn}(\hat{A})$$

$$4 : \left\{ \prod_{t=1}^{|o|} \left| \min \left[ \rho_t(\theta)\hat{A}, \mathrm{clip}(\rho_t(\theta), \epsilon_{\mathrm{low}}, \epsilon_{\mathrm{high}})\hat{A} \right] \right| \right\} \cdot \mathrm{sgn}(\hat{A})$$

$$5 : \left\{ \prod_{t=1}^{|o|} \left| \min \left[ \rho_t(\theta)\hat{A}, \mathrm{clip}(\rho_t(\theta), \epsilon_{\mathrm{low}}, \epsilon_{\mathrm{high}})\hat{A} \right] \right| \right\}^{\frac{1}{|o|}} \cdot \mathrm{sgn}(\hat{A})$$

| Training objectives | AIME24 | AMC | MATH500 | Minerva | Oly. | Avg. |
|---|---|---|---|---|---|---|
| *0* (Pre-RL model) | 16.7 | 38.6 | 50.6 | 9.9 | 16.6 | 26.5 |
| *1* (GRPO) | 40.0 | 59.0 | 83.4 | 32.4 | 41.3 | 51.2 |
| *2* (*without* clip) | 40.0 | 63.9 | 80.6 | 33.5 | 43.7 | 52.3 |
| *3* (*with* seq-clip) | 46.6 | 57.8 | 80.2 | 34.2 | 44.3 | 52.6 |
| *4* (*without* norm) | 36.6 | 67.4 | 82.0 | 29.8 | 44.1 | 52.0 |
| *5* (GMPO) | 43.3 | 61.4 | 82.0 | 33.5 | 43.6 | **52.7** |

**Agentic tasks.** Using Qwen2.5-Instruct-1.5B as the base model, GMPO achieves a 13.1% performance gain over GRPO on ALFWorld, demonstrating its potential in open-world agentic tasks.

### 4.3 ABLATION STUDIES

Table 3 presents an ablation study of the key modifications in GMPO relative to GRPO. The effect of the clipping thresholds is presented in Table 4, and training statistics are shown in Figure 4.

**Geometric mean** *vs*. **Arithmetic mean.** The performance of GRPO and GMPO is reported in lines *1* and *5* of Table 3, respectively. GRPO achieves an average performance of 51.2% by optimizing the arithmetic mean of token-level rewards. In contrast, GMPO improves this to 52.7%, outperforming GRPO by 1.5%, by optimizing the geometric mean instead. In row *4* of Table 3, we test removing the normalization term "$\frac{1}{|o|}$" from the training objective, similar to Dr. GRPO (Liu et al., 2025). This results in a 0.7% drop in average performance (52.0% *vs*. 52.7%), suggesting that the normalization term is crucial for maintaining optimal performance.

**Clipping strategy.** The performance of GMPO without clipping, with token-level clipping, and with sequence-level clipping is shown in lines *2*, *3*, and *5*, respectively. The corresponding ranges of

Table 4: Influence of the clipping thresholds on model performance.

|  | Clipping thresholds $(\epsilon_{\text{low}}, \epsilon_{\text{high}})$ | AIME24 | AMC | MATH500 | Minerva | Oly. | **Avg.** |
|---|---|---|---|---|---|---|---|
| 1 | $(e^{-0.2}, e^{0.2})$ | 36.6 | 60.2 | 84.2 | 35.7 | 45.0 | 52.4 |
| 2 | $(e^{-0.4}, e^{0.4})$ | 43.3 | 61.4 | 82.0 | 33.5 | 43.6 | **52.7** |
| 3 | $(e^{-0.8}, e^{0.8})$ | 40.0 | 60.2 | 82.2 | 33.5 | 44.7 | 52.1 |
| 4 | $(-\infty, +\infty)$ | 40.0 | 63.9 | 80.6 | 33.5 | 43.7 | 52.3 |

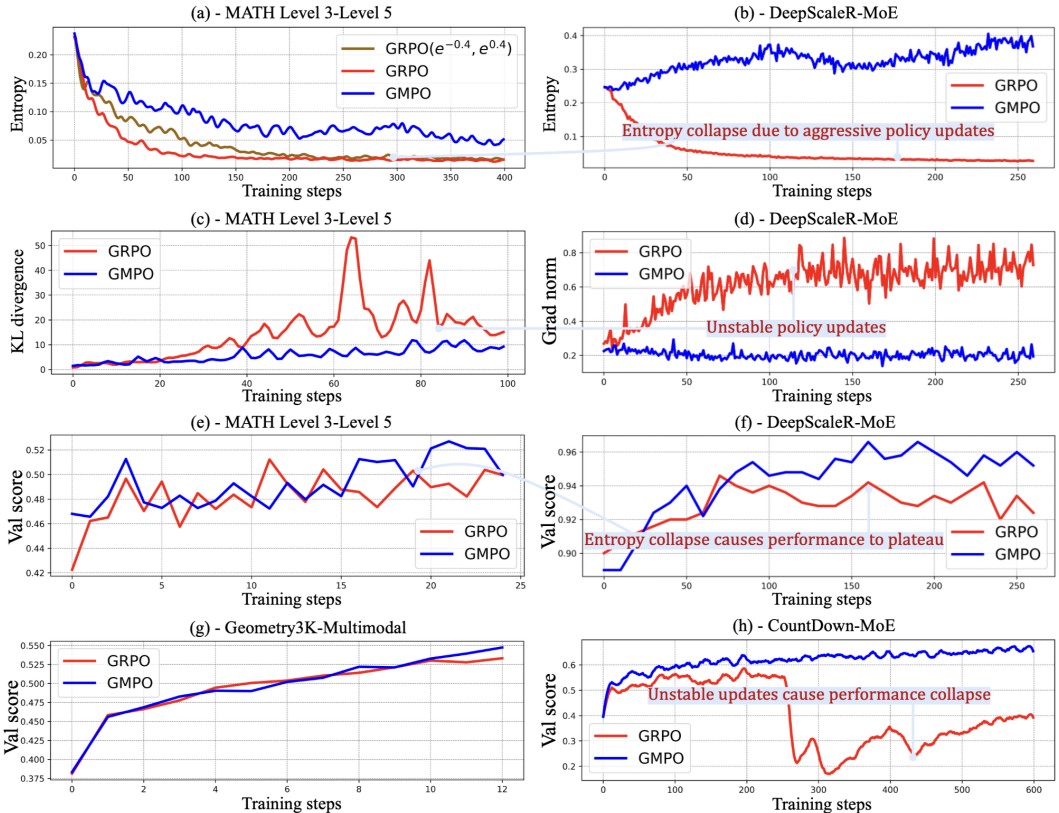

Figure 4: Analysis of entropy, KL divergence, gradient norm, validation score over training steps. (a–b) GMPO maintains higher entropy than GRPO, whether trained on MATH Level 3–Level 5 or DeepScaleR dataset. (c-d) GMPO maintains more stable gradient and a smaller KL divergence from the pre-RL model than GRPO. (e–h) GMPO outperforms GRPO in validation scores across language-only and multimodal tasks, for both dense and Mixture-of-Experts models.

importance sampling ratios are shown in Figure 3, labeled as GMPO $(e^{-0.4}, e^{0.4})$, GMPO-seq-clip-$(e^{-0.4}, e^{0.4})$, and GRPO$(0.8, 1.2)$. Clipping at the sequence-level achieves similar performance to token-level clipping but results in a larger range of importance sampling ratios. Therefore, we use the token-level clipping strategy. Removing the clipping range term (GMPO $(-\infty, +\infty)$) leads to excessive fluctuations in the importance sampling ratio during training, which affects stability and results in a 0.4% decrease in average performance (52.3% vs. 52.7%).

**Influence of the clipping thresholds.** To find the optimal clipping thresholds for GMPO, we train the model with different clipping thresholds, as shown in Table 4 and Figure 3. A larger clipping range encourages exploration but introduces instability to optimization, which ultimately affects performance. To balance stability and performance, we set $(\epsilon_{\text{low}}, \epsilon_{\text{high}})$ in Equation 5 to $(e^{-0.4}, e^{0.4})$, which has a stable range of importance sampling ratios and achieves the best performance.

**Exploration capability.** As noted in (Cui et al., 2025b), language models in reinforcement learning often trade off entropy for short-term performance. Premature entropy collapse can cause performance

to plateau. As shown in Figure 4 (a-b), we visualize the mean token entropy of GMPO and GRPO when training the policy model at MATH Level 3-Level 5 and the more challenging mathematical dataset DeepScaleR. GRPO's entropy drops rapidly during training, limiting exploration and causing performance to plateau (Figure 4 (e–g)). As shown in Figure 4 (a), applying a wider clipping range for GRPO temporarily encourages exploration, but the entropy still declines quickly over time. This behavior arises because GRPO optimizes the arithmetic mean of token-level rewards, which is sensitive to outliers. Consequently, it can generate aggressive updates that sharply reduce entropy while offering only marginal performance gains, hindering both exploration and scalability.

In contrast, GMPO employs the geometric mean, which is robust to outliers. This allows it to maintain stable, moderate entropy, enabling consistent exploration throughout training and resulting in higher rewards and better overall performance than GRPO, as shown in Figure 4 (e–g).

**Training stability.** As shown in Figure 4 (c-d), we visualize the gradient norm during training and the KL divergence between the current model $\pi_\theta$ and the reference model $\pi_{\text{ref}}$. $\pi_{\text{ref}}$ is initialized as the base model before RL training. As training progresses, GMPO maintains a stable gradient and a low KL divergence from the reference model, indicating greater training stability and a lower risk of overfitting. In contrast, GRPO exhibits unstable gradients and large KL divergence, suggesting unstable learning and a greater tendency to drift away from the reference model.

**Validation scores.** Figure 4 (e–h) shows validation scores under different training settings. Figures (e) and (f, g) correspond to Tables 1, while results on CountDown are detailed in Appendix E. GMPO consistently outperforms GRPO in validation scores across language-only (e, f, h) and multimodal (g) tasks, for both dense (e, g) and Mixture-of-Experts models (f, h).

## 5 CONCLUSION

We propose GMPO, a stabilized variant of GRPO. By optimizing the geometric mean of token-level rewards and enlarging the clipping range of the importance sampling ratio, GMPO not only alleviates instability in policy updates but also enhances exploration capabilities, as evidenced by a narrower objective value range, more stable gradients, and consistently lower KL divergence and higher token entropy throughout training. Extensive experiments on language-only and multimodal reasoning benchmarks demonstrate that GMPO outperforms GRPO in terms of both stability and reasoning capacity. This work sets the stage for future research on developing more reliable and scalable reinforcement learning systems.

## 6 ACKNOWLEDGMENT

This work was supported by the National Natural Science Foundation of China (NSFC) under Grant 62472402, 62225208 and 62176260, CAS Project for Young Scientists in Basic Research under Grant No.YSBR-117.

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

# Appendices

## A  GRADIENT DERIVATION

To better understand why GMPO is more stable than GRPO, we analyze its robustness to tokens with extreme importance sampling ratios from a gradient perspective. Specifically, we first derive the gradient of the importance sampling ratio $\rho_{i,t}(\theta)$ with respect to the model parameter $\theta$ in **Lemma 1**. Building on this result, we then derive the gradients of GRPO and GMPO with respect to $\theta$ in **Lemmas 2** and **3**. For clarity, the clipping range term is omitted in the gradient derivation.

**Lemma: 1.** *The derivative of the importance sampling ratio is:*

$$\nabla_\theta \rho_{i,t}(\theta) = \rho_{i,t}(\theta) \nabla_\theta \log(\pi_\theta(o_{i,t}|q, o_{i,<t})) \tag{8}$$

*Proof.*

$$
\begin{aligned}
\nabla_\theta \rho_{i,t}(\theta) &= \nabla_\theta \frac{\pi_\theta(o_{i,t}|q, o_{i,<t})}{\pi_{\theta_{\text{old}}}(o_{i,t}|q, o_{i,<t})} \\
&= \frac{1}{\pi_{\theta_{\text{old}}}(o_{i,t}|q, o_{i,<t})} \nabla_\theta \pi_\theta(o_{i,t}|q, o_{i,<t}) \\
&= \frac{\pi_\theta(o_{i,t}|q, o_{i,<t})}{\pi_{\theta_{\text{old}}}(o_{i,t}|q, o_{i,<t})} \cdot \frac{1}{\pi_\theta(o_{i,t}|q, o_{i,<t})} \cdot \nabla_\theta \pi_\theta(o_{i,t}|q, o_{i,<t}) \\
&= \rho_{i,t}(\theta) \nabla_\theta \log(\pi_\theta(o_{i,t}|q, o_{i,<t}))
\end{aligned}
$$

$\square$

**Lemma: 2.** *The derivative of the GRPO objective is:*

$$\nabla_\theta \mathcal{J}^*_{\text{GRPO}}(\pi_\theta)\Big|_{q,o_i} = \frac{1}{G \cdot |o_i|} \sum_{t=1}^{|o_i|} \rho_{i,t}(\theta) \cdot \hat{A}_i \cdot \nabla_\theta \log(\pi_\theta(o_{i,t}|q, o_{i,<t})) \tag{9}$$

*Proof.*

$$
\begin{aligned}
\nabla_\theta \mathcal{J}^*_{\text{GRPO}}(\pi_\theta)\Big|_{q,o_i} &= \nabla_\theta \frac{1}{G} \sum_{i=1}^{G} \frac{1}{|o_i|} \sum_{t=1}^{|o_i|} \rho_{i,t}(\theta) \hat{A}_i \\
&= \frac{1}{G \cdot |o_i|} \sum_{t=1}^{|o_i|} \cdot \hat{A}_i \cdot \nabla_\theta \rho_{i,t}(\theta) \\
&= \frac{1}{G \cdot |o_i|} \sum_{t=1}^{|o_i|} \rho_{i,t}(\theta) \cdot \hat{A}_i \cdot \nabla_\theta \log(\pi_\theta(o_{i,t}|q, o_{i,<t}))
\end{aligned}
$$

$\square$

**Lemma: 3.** *The derivative of the GMPO objective is:*

$$\nabla_\theta \mathcal{J}^*_{\text{GMPO}}(\pi_\theta)\Big|_{q,o_i} = \frac{1}{G \cdot |o_i|} \sum_{k=1}^{|o_i|} \Big(\prod_{t=1}^{|o_i|} \rho_{i,t}(\theta)\Big)^{\frac{1}{|o_i|}} \cdot \hat{A}_i \cdot \nabla_\theta \log(\pi_\theta(o_{i,k}|q, o_{i,<k})) \tag{10}$$

*Proof.*

$$
\begin{aligned}
\nabla_\theta \mathcal{J}^*_{\text{GMPO}}(\pi_\theta)\Big|_{q,o_i} &= \nabla_\theta \frac{1}{G} \sum_{i=1}^{G} \Big( \prod_{t=1}^{|o_i|} |\rho_{i,t}(\theta)\hat{A}_i| \Big)^{\frac{1}{|o_i|}} \cdot \text{sgn}(\hat{A}_i) \\
&= \nabla_\theta \frac{1}{G} \sum_{i=1}^{G} \Big( \prod_{t=1}^{|o_i|} \rho_{i,t}(\theta) \Big)^{\frac{1}{|o_i|}} \cdot \hat{A}_i \\
&= \frac{1}{G \cdot |o_i|} \Big( \prod_{t=1}^{|o_i|} \rho_{i,t}(\theta) \Big)^{\frac{1}{|o_i|}-1} \cdot \hat{A}_i \cdot \nabla_\theta \prod_{t=1}^{|o_i|} \rho_{i,t}(\theta) \\
&= \frac{1}{G \cdot |o_i|} \Big( \prod_{t=1}^{|o_i|} \rho_{i,t}(\theta) \Big)^{\frac{1}{|o_i|}-1} \cdot \hat{A}_i \cdot \sum_{k=1}^{|o_i|} \Big( \prod_{t=1,t\neq k}^{|o_i|} \rho_{i,t}(\theta) \Big) \nabla_\theta \rho_{i,k}(\theta) \\
&= \frac{1}{G \cdot |o_i|} \Big( \prod_{t=1}^{|o_i|} \rho_{i,t}(\theta) \Big)^{\frac{1}{|o_i|}-1} \cdot \hat{A}_i \cdot \sum_{k=1}^{|o_i|} \Big( \prod_{t=1}^{|o_i|} \rho_{i,t}(\theta) \Big) \nabla_\theta \log(\pi_\theta(o_{i,k}|q,o_{i,<k})) \\
&= \frac{1}{G \cdot |o_i|} \sum_{k=1}^{|o_i|} \Big( \prod_{t=1}^{|o_i|} \rho_{i,t}(\theta) \Big)^{\frac{1}{|o_i|}} \cdot \hat{A}_i \cdot \nabla_\theta \log(\pi_\theta(o_{i,k}|q,o_{i,<k}))
\end{aligned}
$$

$\square$

# B  THEORETICAL CONNECTIONS WITH GRPO

As the foundation of PPO (Schulman et al., 2017) and GRPO Shao et al. (2024), TRPO (Schulman et al., 2015) establishes a monotonic improvement guarantee for general stochastic policies with a trust region constraint. Consequently, under mild assumptions, GRPO inherits these desirable properties from TRPO.

In this section, we show that, within a trust region, GMPO is an $O(\delta^2)$ Lipschitz-stable perturbation of GRPO, where $\delta$ denotes the maximum token-level ratio deviation across the sampled trajectories. As a result, GMPO preserves GRPO's monotonic-improvement and convergence guarantees up to $O(\delta^2)$ error (Bertsekas, 1997), making it a principled optimization objective.

Our argument proceeds in two steps. First, Lemma 4 quantifies the difference between GMPO and GRPO when the policy update remains in a small trust region. Then, Lemma 5 shows that this discrepancy is $O(\delta^2)$, establishing that GMPO is a Lipschitz-stable perturbation of GRPO and therefore inherits its local theoretical guarantees.

**Lemma: 4.** *Let* $\rho_{i,t}(\theta) = \frac{\pi_\theta(o_{i,t}|q,o_{i,<t})}{\pi_{\theta_{\text{old}}}(o_{i,t}|q,o_{i,<t})}$ *be the importance ratio and define the token-level deviation* $\delta_{i,t}(\theta) = \rho_{i,t}(\theta) - 1$. *The trajectory-level mean deviation is* $\delta_i(\theta) = \frac{1}{|o_i|}\sum_{t=1}^{|o_i|}\delta_{i,t}(\theta)$, *and we define the population variance* $\text{Var}_i(\theta) = \frac{1}{|o_i|}\sum_{t=1}^{|o_i|}\big(\delta_{i,t}(\theta) - \delta_i(\theta)\big)^2$. *Assume that the update lies in a trust region where* $|\delta_{i,t}(\theta)| \leq \delta$ *for all* $i,t$ *and some* $\delta > 0$. *Then, as* $\delta \to 0$, *we have*

$$
\mathcal{J}^*_{\text{GMPO}}(\pi_\theta) = \mathcal{J}^*_{\text{GRPO}}(\pi_\theta) - \mathbb{E}\left[ \frac{1}{G} \sum_{i=1}^{G} \frac{\hat{A}_i}{2} \text{Var}_i(\theta) \right] + O(\delta^3). \tag{11}
$$

*Proof.* The training objectives of GRPO ($\mathcal{J}^*_{\text{GRPO}}(\pi_\theta)$) and GMPO ($\mathcal{J}^*_{\text{GMPO}}(\pi_\theta)$) are defined as (clipping range term and KL regularization term are omitted for clarity):

$$
\mathcal{J}^*_{\text{GRPO}}(\pi_\theta) = \mathbb{E}_{q\sim\mathcal{Q},\{o_i\}_{i=1}^G\sim\pi_{\theta_{\text{old}}}(\cdot|q)} \left[ \frac{1}{G} \sum_{i=1}^{G} \frac{1}{|o_i|} \sum_{t=1}^{|o_i|} \rho_{i,t}(\theta)\hat{A}_i \right],
$$

$$
\mathcal{J}^*_{\text{GMPO}}(\pi_\theta) = \mathbb{E}_{q\sim\mathcal{Q},\{o_i\}_{i=1}^G\sim\pi_{\theta_{\text{old}}}(\cdot|q)} \left[ \frac{1}{G} \sum_{i=1}^{G} \Big( \prod_{t=1}^{|o_i|} |\rho_{i,t}(\theta)\hat{A}_i| \Big)^{\frac{1}{|o_i|}} \cdot \text{sgn}(\hat{A}_i) \right].
$$

Then we can deduce the difference between the two objectives as follows:

$$
\mathcal{J}_{\text{GMPO}}^*(\pi_\theta) = \mathbb{E}\left[\frac{1}{G}\sum_{i=1}^{G}\Big(\prod_{t=1}^{|o_i|}|\rho_{i,t}(\theta)\hat{A}_i|\Big)^{\frac{1}{|o_i|}}\cdot\text{sgn}(\hat{A}_i)\right]
$$

$$
= \mathbb{E}\left[\frac{1}{G}\sum_{i=1}^{G}\hat{A}_i\cdot\exp\Big(\frac{1}{|o_i|}\sum_{t=1}^{|o_i|}\log(1+\delta_{i,t}(\theta))\Big)\right]
$$

$$
= \mathbb{E}\left[\frac{1}{G}\sum_{i=1}^{G}\hat{A}_i\cdot\exp\Big(\frac{1}{|o_i|}\sum_{t=1}^{|o_i|}\underbrace{\big[\delta_{i,t}(\theta)-\frac{1}{2}\delta_{i,t}(\theta)^2+O(\delta^3)\big]}_{\text{Taylor expansion of }\log(1+x)}\Big)\right]
$$

$$
= \mathbb{E}\left[\frac{1}{G}\sum_{i=1}^{G}\hat{A}_i\cdot\exp\Big(\delta_i(\theta)-\frac{1}{2}\frac{1}{|o_i|}\sum_{t=1}^{|o_i|}\delta_{i,t}(\theta)^2+O(\delta^3)\Big)\right]
$$

$$
= \mathbb{E}\left[\frac{1}{G}\sum_{i=1}^{G}\hat{A}_i\cdot\underbrace{\Big(1+\delta_i(\theta)-\frac{1}{2}\frac{1}{|o_i|}\sum_{t=1}^{|o_i|}\delta_{i,t}(\theta)^2+\frac{1}{2}\big(\delta_i(\theta)-\frac{1}{2}\frac{1}{|o_i|}\sum_{t=1}^{|o_i|}\delta_{i,t}(\theta)^2\big)^2+O(\delta^3)\Big)}_{\text{Taylor expansion of }\exp(x)}\right]
$$

$$
= \mathbb{E}\left[\frac{1}{G}\sum_{i=1}^{G}\hat{A}_i\cdot\Big(1+\delta_i(\theta)-\frac{1}{2}\text{Var}_i(\theta)-\frac{1}{2}\delta_i(\theta)^2+\frac{1}{2}\delta_i(\theta)^2+O(\delta^3)\Big)\right]
$$

$$
= \mathbb{E}\left[\frac{1}{G}\sum_{i=1}^{G}\hat{A}_i\cdot\Big(1+\delta_i(\theta)-\frac{1}{2}\text{Var}_i(\theta)+O(\delta^3)\Big)\right]
$$

$$
= \mathbb{E}\left[\frac{1}{G}\sum_{i=1}^{G}\frac{1}{|o_i|}\sum_{t=1}^{|o_i|}\rho_{i,t}(\theta)\hat{A}_i\right]-\mathbb{E}\left[\frac{1}{G}\sum_{i=1}^{G}\frac{\hat{A}_i}{2}\text{Var}_i(\theta)\right]+O(\delta^3)
$$

$$
= \mathcal{J}_{\text{GRPO}}^*(\pi_\theta)-\mathbb{E}\left[\frac{1}{G}\sum_{i=1}^{G}\frac{\hat{A}_i}{2}\text{Var}_i(\theta)\right]+O(\delta^3)
$$

which proves the stated relation in Equation 11.

$\square$

**Lemma: 5.** *Assume that: (1) the update lies in a trust region where $|\delta_{i,t}(\theta)|\leq\delta$ for all $i,t$ and some $\delta>0$, (2) the gradient of importance sampling ratio is bounded $||\nabla_\theta\rho_{i,t}(\theta)||\leq K\delta$, (3) the advantage $\hat{A}_i$ is bounded $|\hat{A}_i|\leq A_{\max}$. Then GMPO is an $O(\delta^2)$ Lipschitz-bounded perturbation of GRPO, which means that there exist constants $C_1, C_2>0$ such that for all $\theta$ in the trust region*

$$
\big|\mathcal{J}_{\text{GMPO}}^*(\theta)-\mathcal{J}_{\text{GRPO}}^*(\theta)\big|\leq C_1\,\delta^2, \tag{12}
$$

$$
\big\|\nabla_\theta\mathcal{J}_{\text{GMPO}}^*(\theta)-\nabla_\theta\mathcal{J}_{\text{GRPO}}^*(\theta)\big\|\leq C_2\,\delta^2, \tag{13}
$$

*Proof.* We start from the expansion in Lemma 4:

$$
\mathcal{J}_{\text{GMPO}}^*(\pi_\theta) = \mathcal{J}_{\text{GRPO}}^*(\pi_\theta)-\mathbb{E}\left[\frac{1}{G}\sum_{i=1}^{G}\frac{\hat{A}_i}{2}\,\text{Var}_i(\theta)\right]+O(\delta^3),
$$

where

$$
\text{Var}_i(\theta)=\frac{1}{|o_i|}\sum_{t=1}^{|o_i|}(\delta_{i,t}(\theta)-\delta_i(\theta))^2, \qquad \delta_i(\theta)=\frac{1}{|o_i|}\sum_{t=1}^{|o_i|}\delta_{i,t}(\theta), \qquad \delta_{i,t}(\theta)=\rho_{i,t}(\theta)-1.
$$

Table 5: Comparison of reward aggregators and their performance under same training settings.

| Reward Aggregator | AIME24 | AMC | MATH500 | Minerva | Oly. | Avg. |
|---|---|---|---|---|---|---|
| Interquartile Mean | 36.7 | 60.2 | 79.6 | 29.0 | 43.4 | 49.8 |
| Arithmetic Mean | 40.0 | 59.0 | 83.4 | 32.4 | 41.3 | 51.2 |
| Harmonic Mean | 36.7 | 56.6 | 83.4 | 36.0 | 45.9 | 51.7 |
| Geometric Mean | 43.3 | 61.4 | 82.0 | 33.5 | 43.6 | 52.7 |

Since the update is in a trust region with $|\delta_{i,t}(\theta)| \le \delta$, we can bound the trajectory-level deviation:

$$|\delta_i(\theta)| = \left| \frac{1}{|o_i|} \sum_{t=1}^{|o_i|} \delta_{i,t}(\theta) \right| \le \frac{1}{|o_i|} \sum_{t=1}^{|o_i|} |\delta_{i,t}(\theta)| \le \delta,$$

$$0 \le \mathrm{Var}_i(\theta) = \frac{1}{|o_i|} \sum_{t=1}^{|o_i|} (\delta_{i,t}(\theta) - \delta_i(\theta))^2 \le \frac{1}{|o_i|} \sum_{t=1}^{|o_i|} (|\delta_{i,t}(\theta)| + |\delta_i(\theta)|)^2 \le \frac{1}{|o_i|} \sum_{t=1}^{|o_i|} (2\delta)^2 = 4\delta^2.$$

Hence, the difference in objectives satisfies

$$\left| \mathcal{J}^*_{\mathrm{GMPO}}(\pi_\theta) - \mathcal{J}^*_{\mathrm{GRPO}}(\pi_\theta) \right| = \left| \mathbb{E}\left[ \frac{1}{G} \sum_{i=1}^{G} \frac{\hat{A}_i}{2} \mathrm{Var}_i(\theta) \right] + O(\delta^3) \right| \le 2A_{\max}\delta^2 + O(\delta^3).$$

which proves Equation 12. To prove Equation 13, we differentiate the expansion:

$$\nabla_\theta \mathcal{J}^*_{\mathrm{GMPO}}(\pi_\theta) = \nabla_\theta \mathcal{J}^*_{\mathrm{GRPO}}(\pi_\theta) - \mathbb{E}\left[ \frac{1}{G} \sum_{i=1}^{G} \frac{\hat{A}_i}{2} \nabla_\theta \mathrm{Var}_i(\theta) \right] + O(\delta^2).$$

Now,

$$\nabla_\theta \mathrm{Var}_i(\theta) = \nabla_\theta \frac{1}{|o_i|} \sum_{t=1}^{|o_i|} (\delta_{i,t}(\theta) - \delta_i(\theta))^2$$

$$= \frac{2}{|o_i|} \sum_{t=1}^{|o_i|} (\delta_{i,t}(\theta) - \delta_i(\theta)) \cdot (\nabla_\theta \delta_{i,t}(\theta) - \nabla_\theta \delta_i(\theta)).$$

Since $|\delta_{i,t}(\theta) - \delta_i(\theta)| \le 2\delta$ and $||\nabla_\theta \delta_{i,t}(\theta)|| = ||\nabla_\theta \rho_{i,t}(\theta)|| \le K\delta$, we can bound

$$\|\nabla_\theta \mathrm{Var}_i(\theta)\| \le \frac{2}{|o_i|} \sum_{t=1}^{|o_i|} 2\delta \cdot 2K\delta = 8K\delta^2.$$

Therefore,

$$\left\| \nabla_\theta \mathcal{J}^*_{\mathrm{GMPO}}(\pi_\theta) - \nabla_\theta \mathcal{J}^*_{\mathrm{GRPO}}(\pi_\theta) \right\| \le 4KA_{\max}\delta^2 + O(\delta^2). \tag{14}$$

which proves Equation 13.

$\square$

## C  ADDITIONAL ABLATION STUDIES

As shown in Table 5, we compare the performance and importance sampling ratio ranges of several reward aggregators, including a classic subset of power means (*e.g.*, arithmetic, geometric, and harmonic means) as well as the interquartile mean, formally defined in Definitions 1 and 2.

Among these choices, the geometric mean achieves the strongest overall performance. The arithmetic mean used in GRPO is overly sensitive to large outliers, resulting in unstable importance sampling

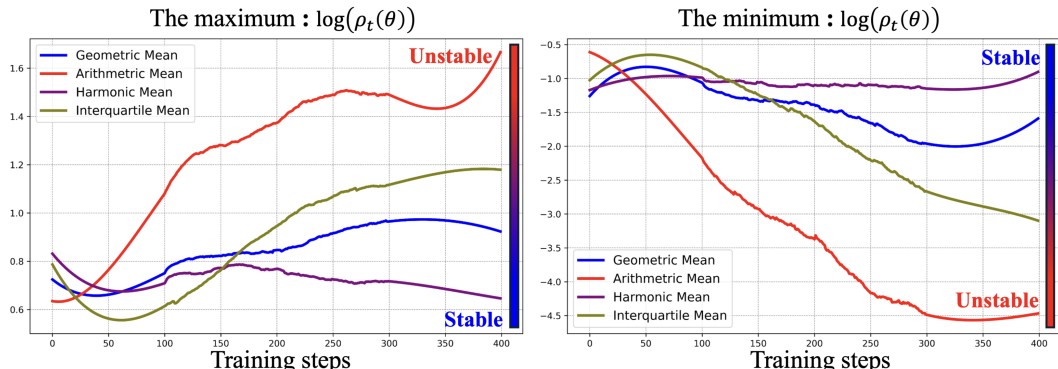

Figure 5: The range of importance sampling ratio $\rho_t(\theta)$ with respect to different reward aggregators and training steps. All curves are smoothed for clarity.

ratios and degraded optimization behavior. The harmonic mean, while yielding the most stable ratios, performs worse than the geometric mean used in GMPO. Finally, the interquartile mean stabilizes the arithmetic mean by filtering outliers. However, its performance falls short of the arithmetic mean, indicating that extreme values might carry meaningful learning signals; overly aggressive trimming removes useful information and, as a result, degrades model performance.

**Definition 1** (Power Mean). *Given non-negative samples $\{x_i\}_{i=1}^n$ and an exponent $p \in \mathbb{R}$, the power mean is defined as*

$$M_p(x_1, \ldots, x_n) = \left( \frac{1}{n} \sum_{i=1}^n x_i^p \right)^{1/p}, \qquad p \neq 0,$$

*and in the limit case*

$$M_0(x_1, \ldots, x_n) = \exp\left( \frac{1}{n} \sum_{i=1}^n \log(x_i) \right).$$

*Special cases include:*

$$M_1 = \text{Arithmetic Mean}, \quad M_0 = \text{Geometric Mean}, \quad M_{-1} = \text{Harmonic Mean}.$$

**Definition 2** (Interquartile Mean). *Given a sample $\{x_i\}_{i=1}^n$ with order statistics $x_{(1)} \leq \cdots \leq x_{(n)}$, the* interquartile mean (IQM) *is the trimmed mean that removes the lowest $25\%$ and highest $25\%$ of the sample:*

$$\text{IQM} = \frac{1}{\lfloor 0.75n \rfloor - \lceil 0.25n \rceil + 1} \sum_{i=\lceil 0.25n \rceil}^{\lfloor 0.75n \rfloor} x_{(i)}.$$

## D  PERFORMANCE ON AGENTIC REINFORCEMENT LEARNING TASKS

To better demonstrate the stability advantages of GMPO over GRPO, we conduct post-training experiments on agentic reinforcement learning (RL) tasks. Following the experimental settings of GiGPO (Feng et al., 2025) and utilizing Qwen2.5-1.5B-Instruct (Qwen et al., 2025) as the base model, we train and evaluate LLM agents on ALFWorld (Shridhar et al., 2020). ALFWorld is an embodied environment designed to assess multi-step decision-making, featuring 3,827 task instances across six common household activity categories: Pick & Place (Pick), Examine in Light (Look), Clean & Place (Clean), Heat & Place (Heat), Cool & Place (Cool), and Pick Two & Place (Pick2). All RL training methods, including our method and baselines, use identical hyperparameter configurations sourced from the verl-agents repo (Feng et al., 2025).

As shown in Table 6, GMPO achieves a significant 13.1% performance gain over GRPO on ALFWorld. Furthermore, GMPO demonstrates comparable performance even when compared to GiGPO, a method specifically designed for agentic RL tasks.

Table 6: Comparison of GMPO and state-of-the-art methods on mathematical reasoning benchmarks.

| Agentic Model | Pick | Look | Clean | Heat | Cool | Pick2 | ALL |
|---|---|---|---|---|---|---|---|
| *Closed-Source Model* | | | | | | | |
| GPT-4o | 75.3 | 60.8 | 31.2 | 56.7 | 21.6 | 49.8 | 48.0 |
| Gemini-2.5-Pro | 92.8 | 63.3 | 62.1 | 69.0 | 26.6 | 58.7 | 60.3 |
| *Open-Source Model* | | | | | | | |
| Qwen2.5-1.5B-Instruct Qwen et al. (2025) | 5.9 | 5.5 | 3.3 | 9.7 | 4.2 | 0.0 | 4.1 |
| PPO-1.5B Schulman et al. (2017) | 64.8 | 40.5 | 57.1 | 60.6 | 46.4 | 47.4 | 54.4 |
| RLOO-1.5B Ahmadian et al. (2024) | 88.3 | 52.8 | 71.0 | 62.8 | 66.4 | 56.9 | 69.7 |
| GRPO-1.5B Shao et al. (2024) | 85.3 | 53.7 | 84.5 | 78.2 | 59.7 | 53.5 | 72.8 |
| GiGPO-1.5B Feng et al. (2025) | 94.4 | 67.5 | 94.8 | 94.4 | 79.8 | 76.4 | **86.7** |
| GMPO-1.5B (Ours) | 93.1 | 78.6 | 81.0 | 88.2 | 82.1 | 89.5 | 85.9 |

Table 7: Training settings for GMPO and GRPO on Mixture-of-Experts models. Qwen2.5-200M[†] is a small-scale language model adapted from the Qwen2.5 series (Qwen et al., 2025). "Bs / Mini Bs" denote the batch size and mini-batch size used during training, respectively. "E./Act. E." indicate the total number of experts in the model and the number of experts activated per token, respectively.

| Training dataset | Eval dataset | Base model | Bs./Mini Bs. | E./Act. E. |
|---|---|---|---|---|
| DeepScaleR | MATH500 | Qwen3-32B Yang et al. (2025) | 128/64 | 128/8 |
| CountDown | CountDown(Val) | Qwen2.5-200M[†] Bai et al. (2025) | 256/128 | 8/1 |

# E    PERFORMANCE ON MIXTURE-OF-EXPERTS MODELS

To better demonstrate the stability advantage of GMPO over GRPO, we conduct post-training experiments on Mixture-of-Experts (MoE) models, where stability is particularly critical. The experiments are performed on the DeepScaleR (Luo et al., 2025) and CountDown (Pan, 2024) datasets, with detailed training settings provided in Table 7. Specifically, DeepScaleR consists of approximately 40,000 unique mathematics problem-answer pairs compiled from AIME (Li et al., 2024), AMC (Li et al., 2024), Omni-MATH dataset, and Still dataset. CountDown consists of arithmetic puzzles where models combine given numbers using basic operations to reach a target, commonly used to test algorithmic reasoning and step-by-step problem solving. We reserve a subset of the CountDown dataset for model evaluation.

**CountDown.** As shown in Figure 6(a)(c), we visualize the KL divergence and gradient norm during GMPO and GRPO training. GMPO consistently maintains a lower KL divergence from the reference model and a steadier gradient norm than GRPO. Consequently, GMPO achieves stable validation scores, whereas GRPO collapses after about 250 steps, as shown in Figure 6 (e).

**DeepScaleR.** As shown in Figure 6 (b)(d), GMPO achieves higher entropy while a steadier gradient norm than GRPO. Consequently, GMPO achieves higher validation scores as shown in Figure 6 (f).

# F    ANALYSIS OF THE NORMALIZATION FACTOR IN THE GEOMETRIC-MEAN

Unlike vanilla GRPO in DeepSeek-math (Shao et al., 2024), DeepSeek-R1 (Guo et al., 2025a) maximizes the sequence-level reward $(\prod_{t=1}^{|o_i|} \rho_{i,t}(\theta))\hat{A}_i$. The term $\prod_{t=1}^{|o_i|} \rho_{i,t}(\theta)$ also appears in the objective of GMPO (Equation 5). Unlike DeepSeek-R1, GMPO introduces an additional power-based normalization term: "$\frac{1}{|o_i|}$", which we find is critical for GMPO objective. As shown in Figure 7, we visualize the range of sequence-level importance sampling ratios from trajectories that yield positive rewards during GRPO training. Without the normalization term, these sequence-level importance sampling ratios can become very large, especially as the response length increases. This phenomenon ultimately leads to unstable policy optimization, which in turn degrades the model's final performance.

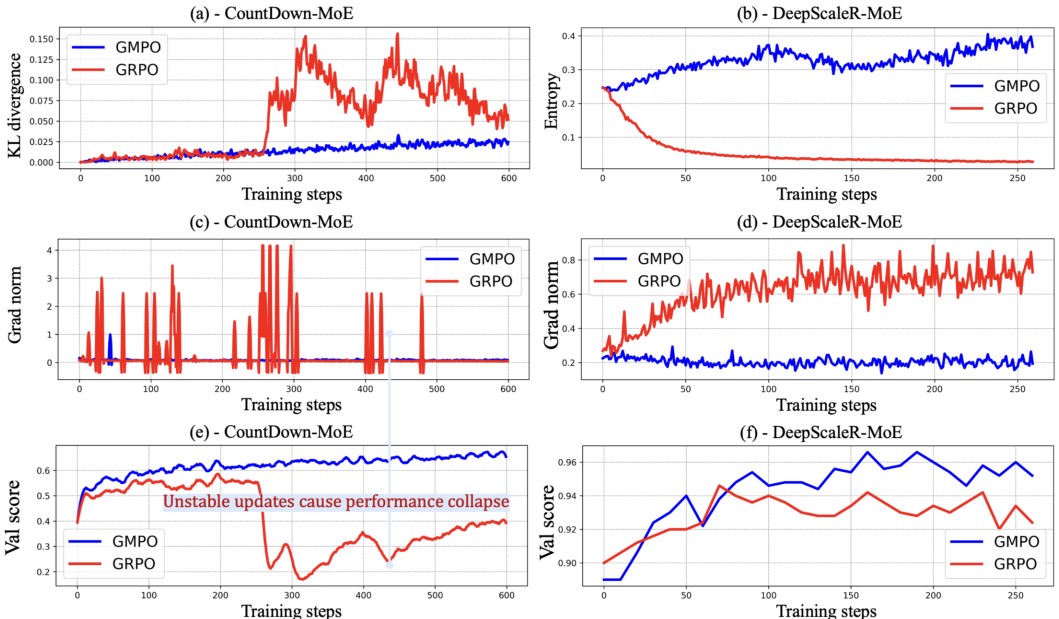

Figure 6: Analysis of entropy, KL divergence, gradient norm, and validation score over training steps on Mixture-of-Experts models. (a) GMPO maintains smaller KL divergence than GRPO. (b) GMPO maintains higher entropy than GRPO. (c-d) GMPO maintains more stable gradient norm than GRPO, suggesting more stable policy optimization. (e-f) GMPO achieves higher validation score than GRPO.

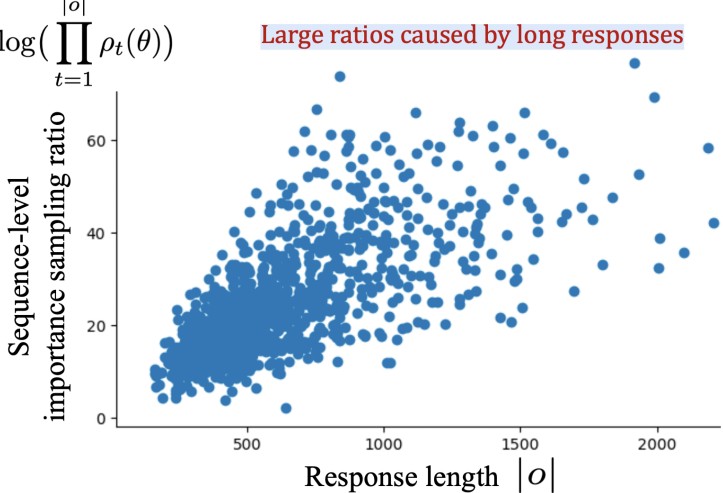

Figure 7: Sequence-level importance sampling ratios from trajectories that yield positive rewards during GRPO training. Without normalization, these ratios can become highly unstable, especially as the response length increases.

## G    USE OF LARGE LANGUAGE MODELS

During the writing process, we consulted large language models (LLMs) for word choice suggestions to enhance readability. The final manuscript was carefully reviewed by humans to prevent any potential inaccuracies or misleading information generated by the LLMs.

