# OpenReview forum: "Geometric-Mean Policy Optimization"
_ICLR.cc/2026/Conference — ICLR 2026 Poster_

### Official Review · Reviewer_WSoW · 2025-10-18

**Soundness:** 3
**Presentation:** 3
**Contribution:** 2
**Rating:** 2
**Confidence:** 4

**Summary:**

This paper proposes Geometric-Mean Policy Optimization (GMPO), a simple and effective variant of GRPO for stabilizing RL-based fine-tuning of LLMs. While GRPO maximizes the arithmetic mean of token-level importance-weighted rewards, GMPO instead maximizes its geometric mean, which is theoretically less sensitive to outliers.

The paper provides:

- A clear analysis showing that GMPO has a narrower objective range and smaller variance in importance sampling ratios.
- A gradient-based justification showing geometric averaging provides smoother updates.
- Extensive experiments on reasoning benchmarks (AIME24, AMC, MATH500, Minerva, OlympiadBench, Geometry3K, and MoE settings).
GMPO improves average Pass@1 by 4.1% on GRPO-7B and shows improved entropy stability and lower KL divergence during training.

The paper is clearly written and supported by sufficient experiments. However, I have several reservations about its theoretical depth and novelty (see the weaknesses section). In particular, the connection between the proposed theory and the empirical behavior of the model remains somewhat unclear. Additionally, the idea is closely related to **Group Sequence Policy Optimization (GSPO)**, which seems to share a similar motivation and formulation. Depending on how ICLR treats concurrent submissions and the arXiv paper, this overlap may further reduce the perceived novelty. Given these concerns, I lean toward a negative recommendation at this stage.

**Strengths:**

1. **Simple but intuitive idea.**

    Replacing arithmetic with geometric mean provides a clear and theoretically motivated way to suppress reward outliers, which is an elegant modification of GRPO with minimal implementation overhead.

2. **Extensive experiments.**

    Results across multiple reasoning and multimodal benchmarks (including MoE settings) are consistent and demonstrate improvement without introducing extra hyperparameters or architectural changes

3. **Comprehensive ablation.**

    The paper carefully studies token-level vs. sequence-level clipping, clipping range, normalization, and entropy behavior, which strengthens empirical credibility.

4. **Readable and well-structured.**

    The presentation is clear, the pseudo-code and figures are well-designed, and the comparisons with Dr.GRPO, DAPO, and GRPO variants are complete.

**Weaknesses:**

1. **Limited conceptual novelty.**

    While the geometric mean idea is intuitive and effective, it is arguably a *minor variation* of GRPO rather than a fundamentally new algorithmic contribution. The theoretical part mainly formalizes this intuition but lacks deeper insight into the connection to existing stability mechanisms (e.g., adaptive learning rate, variance regularization). The existence of GSPO further weakens the novelty of this paper.

2. **The analysis is not quite clear to some extent.** The main results of analyzing the gradient space of GMPO and GRPO lie in equations 5 and 6, where we see that the main difference is the reweighting coefficient of the gradient to log-pi. Then, how do the results relate to the model’s improved behaviors in:

    - The more stabilized dynamics are shown in Figure 3. One potential explanation is the insensitivity to the outliers of the importance sampling ratio; then, would vanilla GRPO+filter on this ratio solve the problem? If not, where are the benefits of applying the geometric mean here?

    - Figure 4 provides many good signs that GMPO indeed stabilizes the model’s training. Could the authors provide more analysis on why GMPO can make them happen, e.g., why the entropy is not decaying, and why the gradient norm is more stable? I believe that only claiming to the insensitive to the outlier is not enough.

**Questions:**

1. Importance sampling, not important sampling.

2. This is a major concern. In Section 3 (line 262), the authors state that GMPO performs clipping at the token level, which is different from GRPO. However, when comparing Equations (4) and (1), the clipping operations appear to be applied inside the summation (or product) over the token index t. Furthermore, by examining the gradients in Equations (5) and (6), we can see that GRPO’s rho depends explicitly on t, whereas GMPO’s weight is normalized across all t.

    In other words, GRPO effectively operates in a token-level manner, while GMPO behaves more like a sequence-level method, which contradicts the authors’ claim that GMPO’s token-level behavior.

---

> ### Author Response · Authors · 2025-11-21
> **Response to WSoW (Part 1/3)**
>
> **Q1: The idea is closely related to Group Sequence Policy Optimization (GSPO), which seems to share a similar motivation and formulation. Depending on how ICLR treats concurrent submissions and the arXiv paper, this overlap may further reduce the perceived novelty.**
>
> **R1:** Questioning the novelty of this work by comparing it with GSPO is inappropriate for the following reasons:
>
> 1. **Independent and concurrent work:** GMPO and GSPO are concurrent works conducted independently.
>
> 2. **Unpublished status:** GSPO is an unpublished arXiv paper.
>
> 3. **Distinct motivations, objectives, and experiments:** The two methods differ substantially in all aspects. Specifically:
>
>    - **Motivation:** GSPO transforms the GRPO into a sequence-level variant, while GMPO replaces the reward normalization with a geometric mean.
>
>    - **Objective function:** GSPO employs a sequence-level clipping strategy, making it a sequence-level RL algorithm, while GMPO adopts token-level clipping, making it a token-level RL algorithm.
>
>    - **Experiments:** GSPO focuses primarily on MoE models, whereas GMPO conducts extensive experiments across multiple model sizes (1.5B, 7B, and 32B MoE), different architectures (dense and MoE), and diverse domains (language-only, multimodal reasoning, agentic RL).
>
> **Q2: While the geometric mean idea is intuitive and effective, it is arguably a minor variation of GRPO rather than a fundamentally new algorithmic contribution.**
>
> **R2:** *Evaluating an algorithm’s contribution solely based on the magnitude of the change is not appropriate*. The geometric mean idea is simple-yet-effective, and we have provided both theoretical and empirical evidence to show that the modification works and generalizes across different settings. If the reviewer can point us to relevant literature on existing stability mechanisms, we would be happy to provide a more detailed theoretical comparison.
>
> **Q3: The main results of analyzing the gradient space of GMPO and GRPO lie in equations 5 and 6, where we see that the main difference is the reweighting coefficient of the gradient to log-pi. Then, how do the results relate to the model’s improved behaviors in …**
>
> **R3:** **From a gradient perspective, GMPO is more stable than GRPO**. Specifically, we explain why GMPO is more stable than GRPO from a gradient perspective in lines 214-266 in the paper:
>
> Equation 5: ${\\displaystyle \\nabla _\\theta} \\mathcal{J}^{*} _\\mathrm{GRPO}(\\pi _\\theta) \\Big| _{q, o_i} = \\frac{1}{G \\cdot |o_i|} \\sum _{t=1}^{|o_i|} {\\color{red}\\rho _{i,t}(\\theta)} \\cdot \\hat{A_i} \\cdot {\\displaystyle \\nabla _\\theta} \\mathrm{log} (\\pi _\\theta (o _{i,t} | q, o _{i,<t}))$
>
> Equation 6: ${\\displaystyle \\nabla _\\theta} \\mathcal{J}^{*} _\\mathrm{GMPO}(\\pi _\\theta) \\Big| _{q, o_i} = \\frac{1}{G \\cdot |o_i|} \\sum _{t=1}^{|o_i|} {\\color{blue}\\Big( \\prod _{k=1}^{|o_i|} \\rho _{i,k}(\\theta) \\Big)^{\\frac{1}{|o_i|}}} \\cdot \\hat{A_i} \\cdot {\\displaystyle \\nabla _\\theta} \\mathrm{log} (\\pi _\\theta (o _{i,t} | q, o _{i,<t}))$
>
> ``*The gradients of both objectives are weighted sums of the policy gradients of the generated tokens, but with different weights. For $\mathcal{J}^{\*} _\mathrm{GRPO}(\pi _{\theta})$, the weight of the token $o _{i,t}$ includes its individual importance sampling ratio $\rho _{i,t}(\theta)$. An extreme $\rho _{i,t}(\theta)$ will cause the token gradient to be too large or small, resulting in aggressive policy updates. For $\mathcal{J}^{\*} _{\mathrm{GMPO}}(\pi _{\theta})$, the weight of the token $o _{i,t}$ includes the geometric mean of all the ratios $\big(\prod _{k=1}^{|o_i|}\rho _{i,k}(\theta)\big)^{\frac{1}{|o_i|}}$ in the same sequence, provide a more balanced update signal and is more robust to outlier values.*''

---

> ### Author Response · Authors · 2025-11-21
> **Response to WSoW (Part 2/3)**
>
> **Q4: would vanilla GRPO+filter on this ratio solve the problem? If not, where are the benefits of applying the geometric mean here?**
>
> **R4:** *GRPO+filter cannot resolve the stability issue*. Specifically, for all GRPO experiments reported in this paper, the default token-level clipping strategy was used to filter extreme importance sampling ratios. Despite the use of strict filtering thresholds, GRPO still exhibits high variance in its importance sampling ratios (Figure 3 in the paper). This instability is even more pronounced in MoE models, where GRPO can experience sudden performance collapses during training (Figure 4(h) in the paper). By contrast, training with the geometric mean significantly reduces the variance of the importance sampling ratios under the same clipping thresholds, thereby improving training stability (Figures 1 and 3 in the paper).
>
>
>
> **Q5: Could the authors provide more analysis on why GMPO can make them happen, e.g., why the entropy is not decaying.**
>
> **R5:** Thank you for the suggestion. We have added further explanation on why GMPO prevents entropy decay. Specifically, the entropy collapse under the arithmetic mean arises because it assigns excessively large gradients to tokens whose probabilities are positively correlated with their advantages (these tokens have importance sampling ratios $\rho _{i,t}(\theta) > 1$ and advantages $\hat{A_i} > 0$).
>
> According to Theorem 2 from the entropy mechanism [1]:
> $$\\mathcal{H} \\Big(\\pi _{\\theta}^{k+1} \\mid s \\Big) - \\mathcal{H} \\Big(\\pi _{\\theta}^{k} \\mid s \\Big) \\approx -\\eta \\cdot \\mathrm{Cov} _{a \\sim \\pi _{\\theta}^{k}(\\cdot \\mid s)} \\Big( \\log \\pi _{\\theta}^{k}(a \\mid s), A(s,a) \\Big)$$
> the rate of entropy decay (left term) is directly tied to this covariance (right term). *Because GRPO places disproportionately high weight on these high-covariance tokens, its entropy collapses rapidly during training. In contrast, the geometric mean suppresses the influence of such tokens, reducing their gradient magnitude and thus slowing entropy decay.*
>
> [1] Cui, Ganqu, et al. "The entropy mechanism of reinforcement learning for reasoning language models." arXiv preprint arXiv:2505.22617 (2025)
>
> **Q6: why the gradient norm is more stable?**
>
> **R6:** In our response to Q3, we explained from the gradient perspective why GMPO is more stable than GRPO. Because GMPO avoids over-weighting the gradients of individual tokens with extremely large importance sampling ratios, it prevents the model from overfitting to a small set of large-gradient tokens. Consequently, GMPO is expected to produce a smaller KL divergence from the base model (i.e., less overfitting) and a more stable gradient norm throughout training.
>
> **Q7: Importance sampling, not important sampling.**
>
> **R7:** Thank you for your suggestion; we have updated the relevant text in the manuscript.

---

> ### Author Response · Authors · 2025-11-21
> **Response to WSoW (Part 3/3)**
>
> **Q8: This is a major concern. In Section 3 (line 262), the authors state that GMPO performs clipping at the token level, which is different from GRPO. However, when comparing Equations (4) and (1), the clipping operations appear to be applied inside the summation (or product) over the token index t.**
>
> **R8:** In Section 3 (line 262), our original text states: “Unlike the vanilla GRPO in DeepSeek-math [1], DeepSeek-R1[2] maximizes the sequence-level reward $(\prod_{t=1}^{|o_i|}\rho_{i,t}(\theta))\hat{A_i}$ and applies clipping at the sequence level, i.e., $\mathrm{clip}\big(\prod_{t=1}^{|o_i|}\rho_{i,t}(\theta),\epsilon_\mathrm{low},\epsilon_\mathrm{high}\big)$.”  This means that the **the GRPO versions in DeepSeek-math and DeepSeek-R1 are different**:
>
> DeepSeek-math adopts a token-level GRPO algorithm with token-level clipping:
>
> $\\mathcal{J} _{GRPO-DeepSeek-math}(\\theta) = \\mathbb{E} _{[q \\sim P(Q), \\{o_i\\} _{i=1}^G \\sim \\pi _{\\theta _{old}}(O | q)]}
>  \\frac{1}{G} \\sum _{i=1}^G \\frac{1}{|o_i|} \\sum _{t=1}^{|o_i|} \Big( \\min \Big( \\frac{\\pi _\\theta(o _{i, t} |q, o _{i, <t})}{\\pi _{\\theta _{old}}(o _{i, t} |q, o _{i, <t})} A _{i,t}, \text{clip} \Big( \\frac{\\pi _\\theta(o _{i, t} |q, o _{i, <t})}{\\pi _{\\theta _{old}}(o _{i, t} |q, o _{i, <t})}, 1 - \\epsilon, 1 + \\epsilon \\Big)  A _{i,t} \\Big) - \\beta \\mathbb{D} _{KL} \\Big(\\pi _{\\theta} || \\pi _{ref} \\Big) \\Big),$
>
>  whereas DeepSeek-R1 adopts a sequence-level GRPO algorithm with sequence-level clipping:
>
> $\\mathcal{J} _{GRPO-DeepSeek-R1}(\\theta) = \\mathbb{E} _{[q \\sim P(Q), \\{o_i\\} _{i=1}^G \\sim \\pi _{\\theta _{old}}(O | q)]}
>  \\frac{1}{G} \\sum _{i=1}^G \Big( \\min \Big( \\frac{\\pi _\\theta(o_i |q)}{\\pi _{\\theta _{old}}(o_i |q)} A_i, \text{clip} \Big( \\frac{\\pi _\\theta(o_i |q)}{\\pi _{\\theta _{old}}(o_i |q)}, 1 - \\epsilon, 1 + \\epsilon \\Big)  A_i \\Big) - \\beta \\mathbb{D} _{KL} \\Big(\\pi _{\\theta} || \\pi _{ref} \\Big) \\Big),$
>
> *We did not claim that "GMPO performs clipping at the token-level, which is different from GRPO". Instead, we compared the token-level clipping used in DeepSeek-math with the sequence-level clipping used in DeepSeek-R1, analyzed which is more suitable for GMPO, and ultimately chose the token-level clipping strategy.*
>
> [1] Shao, Zhihong, et al. "Deepseekmath: Pushing the limits of mathematical reasoning in open language models." arXiv preprint arXiv:2402.03300 (2024).
>
> [2] Guo, Daya, et al. "Deepseek-r1: Incentivizing reasoning capability in llms via reinforcement learning." arXiv preprint arXiv:2501.12948 (2025).
>
> **Q9: Furthermore, by examining the gradients in Equations (5) and (6), we can see that GRPO’s rho depends explicitly on t, whereas GMPO’s weight is normalized across all t. In other words, GRPO effectively operates in a token-level manner, while GMPO behaves more like a sequence-level method, which contradicts the authors’ claim that GMPO’s token-level behavior.**
>
> **R9:** *The key reason GMPO should be regarded as a token-level method is that it fundamentally relies on token-level clipping.* In Equation (5) and (6), clipping range term is omitted for clarity.
> Under this condition, GMPO indeed exhibits characteristics similar to a sequence-level method.

---

> > ### Comment · Reviewer_WSoW · 2025-11-22
> >
> > Thanks very much for the author's clarification. Most of my concerns are well addressed. I will raise my evaluation to 6 due to the following reasons:
> > 1. I checked the rule of co-current work for ICLR. Indeed, the existence of GSPO should not harm the novelty of GMPO
> > 2. The theoretical part is significantly improved (in Appendix B and R5)

---

### Official Review · Reviewer_VUr8 · 2025-10-30

**Soundness:** 3
**Presentation:** 3
**Contribution:** 3
**Rating:** 8
**Confidence:** 5

**Summary:**

The paper proposes Geometric-Mean Policy Optimization (GMPO), a modification of Group Relative Policy Optimization (GRPO) designed to improve training stability for large language models. The core issue identified with GRPO is its use of the arithmetic mean of token-level rewards, which makes it sensitive to outliers in importance sampling (IS) ratios and leads to unstable policy updates. GMPO addresses this by replacing the arithmetic mean with the geometric mean, which is inherently more robust to outliers. The authors provide theoretical analysis showing that the GMPO objective has a narrower value range and results in more balanced gradient updates. Empirically, through experiments on several mathematical and multimodal reasoning benchmarks, the paper demonstrates that GMPO consistently outperforms GRPO, achieving significant performance gains (e.g., an average Pass@1 improvement of up to 4.1%). The paper includes extensive ablation studies that validate the key design choices of GMPO, such as token-level clipping and the normalization factor in the geometric mean.

**Strengths:**

1.  The core idea of replacing the arithmetic mean with the geometric mean is simple, elegant, and well-motivated. It directly targets a plausible source of instability in GRPO (sensitivity to outlier rewards) with a classic statistical tool known for its robustness. The "plug-and-play" nature of the modification makes it highly practical.
2.  The empirical evaluation is comprehensive and convincing. The authors demonstrate consistent improvements over GRPO across multiple model sizes (1.5B, 7B, 32B MoE), different model architectures (dense and MoE), and different domains (language-only and multimodal reasoning). The performance gains are substantial and statistically significant.
3.  The ablation studies presented in Section 4.3 are excellent. They systematically dissect the contributions of each component of the proposed method: the geometric mean itself, the token-level clipping strategy, the clipping range, and the normalization term. This analysis provides strong evidence for the authors' design choices and adds depth to the paper's claims.
4.  The paper is generally well-written and clearly structured. The motivation for the work is established effectively in the introduction, and the figures, particularly Figure 1 and Figure 4, provide strong visual support for the claims regarding improved stability (tighter IS ratio range, higher entropy, lower KL divergence) and its connection to final performance.

**Weaknesses:**

1.  The manuscript has numerous small but distracting typographical and formatting errors.
    *   The text is missing a character at the beginning of lines 190 and 202.
    *   The formulas in the derivation on lines 182-189 lack eq number.
    *   Several plots in Figure 4 are missing x-axis labels (e.g., plots c, e, g), which should be labeled "Training steps" for clarity.
2.  The related work section, while comprehensive, reads like a long list of recent methods without sufficient structure or synthesis. It does not adequately position GMPO with respect to other methods that also aim to improve stability, such as those using adaptive reward normalization (BNPO) or variance reduction techniques (OPO). The paper would be stronger if it discussed the conceptual differences and potential synergies with these alternative approaches, rather than solely focusing on the direct comparison with GRPO.

**Questions:**

1.  In line 266, it says "clipping at the token-level, as shown in Eq 3". I didn't see the clipping operation in formula 3; should this refer to Eq 4?
2. What is the intended value of $sgn(\hat A_i)$ when $\hat A_i=0$? The current code treats it as negative; the math does not specify it. This edge case can occur because the group based advantage in GRPO is standardized. Please clarify.
3. Can you report GRPO with the exact same per token clipping in log space and the same wide range $(e^{-0.4},e^{0.4})$ so that the only difference is arithmetic vs geometric aggregation? That would isolate the effect of the geometric mean from the effect of wider exploration.
4. In Table 3 you compare against several external methods such as PRIME Zero and OpenReasoner Zero. Were those numbers reproduced under your pipeline or copied from their papers which might have used different data or evaluation protocols?

---

> ### Author Response · Authors · 2025-11-21
> **Response to VUr8 (Part 1/1)**
>
> **Q1: The manuscript has numerous small but distracting typographical and formatting errors.**
>
> **R1:** Thank you for pointing this out. We have thoroughly checked the paper and corrected the errors in the revision.
>
> **Q2: The paper would be stronger if it discussed the conceptual differences and potential synergies with these alternative approaches, rather than solely focusing on the direct comparison with GRPO.**
>
> **R2:** Thank you for the suggestion. We have refined the related work section and strengthened the comparison with relevant methods in lines 140–146. Specifically, we revised the paragraph to:
>
>  ``*Despite rapid progress, the stability of RL for LLMs remains rarely explored, even though it is essential for developing reliable and scalable post-training systems. While several GRPO variants enhance stability through better baseline estimation (OPO), reward shaping (GRPO-lead) or reward normalization (BNPO), the underlying stability of the RL process remains a persistent challenge. Our work offers a complementary perspective on these methods by introducing a robust aggregation operator for token-level rewards, providing an orthogonal approach to achieving more reliable and scalable post-training systems.*’’
>
> **Q3: What is the intended value of when The current code treats it as negative; the math does not
> specify it. This edge case can occur because the group based advantage in GRPO is standardized.
> Please clarify.**
>
> **R3:** When $\hat{A_i}=0$, the first term in the GMPO objective, the gradient of $\sum_{i=1}^{G}\Big\\{\prod_{t=1}^{|o_i|}\Big|\min \big[\rho_{i,t}(\theta)\hat{A_i}, \mathrm{clip}(\rho_{i,t}(\theta), \epsilon_\mathrm{low}, \epsilon_\mathrm{high}) \hat{A_i}\big]\Big|\Big\\}^{\frac{1}{|o_i|}}$ has a zero gradient with respect to $\theta$ (it can also be inferred from Equation 6 in the paper). Therefore, the value of $\mathrm{sgn}(\hat{A_i})$ at $\hat{A_i}=0$ does not affect the gradient of the objective.
>
> **Q4: Can you report GRPO with the exact same per token clipping in log space and the same wide range so that the only difference is arithmetic vs geometric aggregation?**
>
> **R4:** As suggested, we performed additional experiments on clipping ratios. We find that simply enlarging the clipping range in GRPO leads to training instability, causing performance fluctuations or even degradation. In contrast, broader thresholds can improve GMPO results, benefiting from the inherent stability of the geometric mean.
>
> | Model | Clipping thresholds  | AIME24 | AMC  | MATH500 | Minerva | Oly. | **Avg.** |
> |-----|---------------|--------|------|---------|---------|------|----------|
> | GRPO  | $(e^{-0.4}, e^{0.4})$ |  33.3  | 61.4  |  81.6   | 34.6  | 43.4 | 50.9     |
> | GRPO  | (0.8, 1.2)   | 40.0   | 59.0 | 83.4    | 32.4    | 41.3 | 51.2     |
> | GMPO | $(e^{-0.4}, e^{0.4})$     | 43.3   | 61.4 | 82.0    | 33.5    | 43.6 | 52.7     |
>
> **Q5: In Table 3 you compare against several external methods such as PRIME Zero and OpenReasoner Zero. Were those numbers reproduced under your pipeline or copied from their papers which might have used different data or evaluation protocols?**
>
> **R5:** The experimental settings of GMPO, GRPO, and Dr. GRPO [1] in this paper strictly follow those described in Dr. GRPO and its official codebase to ensure fair and consistent comparison. For other results reported in Table 3, such as OpenReasoner-Zero, SimpleRL-Zero-7B, and PRIME-Zero-7B, we directly adopt the numbers reported in Dr.GRPO.
>
> [1] Liu, Zichen, et al. "Understanding r1-zero-like training: A critical perspective." arXiv preprint arXiv:2503.20783 (2025).

---

> > ### Comment · Reviewer_VUr8 · 2025-11-26
> >
> > The authors have satisfactorily addressed all of my concerns. I find the revised paper to be of high quality and I support its acceptance.

---

### Official Review · Reviewer_vzMt · 2025-11-01

**Soundness:** 3
**Presentation:** 3
**Contribution:** 2
**Rating:** 6
**Confidence:** 4

**Summary:**

The paper proposed GMPO, which replaces the arithmetic mean in GRPO with the geometric mean to achieve better  stability of training.

**Strengths:**

The method is easy to follow, and the paper writing is clear.

**Weaknesses:**

* Judging from Table 4, it seems that the proposed method's gain is quite marginal. The token-wise clip also seems not so effective.
* The major technical change from GRPO seems to be token-level clipping + geometric mean vs group arithmetic mean. The technical novelty is low.
* Although discussion is present, comparisons is missing against the method that controls training stability by carefully setting clipping ratio (e.g. DAPO)

**Questions:**

See weakness

---

> ### Author Response · Authors · 2025-11-21
> **Response to vzMt (Part 1/1)**
>
> **Q1: Judging from Table 4, it seems that the proposed method's gain is quite marginal.**
>
> **R1:** We have made a comprehensive comparison with the baseline (GRPO) in Table 1 (Line 324-345 in the paper). Specifically, *GMPO consistently outperforms GRPO across multiple model sizes (1.5B, 7B, 32B), different model architectures (dense and MoE), and different domains (language-only, multimodal reasoning, agentic RL)*, which are far beyond marginal gains. In addition, we also employ GMPO in ALFWorld [1], which is a reasoning benchmark related to tool-use and embodied sequential reasoning. GMPO achieves a significant `13.1%` performance gain over GRPO on ALFWorld. (**More details in Appendix D**)
>
> | Agentic Model                               | Pick  | Look  | Clean | Heat  | Cool  | Pick2 | **ALL** |
> |--------------------------------------------|-------|-------|-------|-------|-------|-------|---------|
> | Qwen2.5-1.5B-Instruct             | 5.9   | 5.5   | 3.3   | 9.7   | 4.2   | 0.0   | 4.1     |
> | GRPO-1.5B                        | 85.3  | 53.7  | 84.5  | 78.2  | 59.7  | 53.5  | 72.8    |
> | GMPO-1.5B (Ours)                       | 93.1  | 78.6  | 81.0  | 88.2  | 82.1  | 89.5  | 85.9    |
>
> [1] Shridhar, Mohit, et al. "ALFWorld: Aligning Text and Embodied Environments for Interactive Learning." International Conference on Learning Representations.
>
>
> **Q2: The token-wise clip also seems not so effective.**
>
> **R2:** **Token-wise clipping is substantially more stable than sequence-wise clipping.** In particular, the variance of the importance sampling ratios under token-wise clipping is much lower than that of sequence-wise clipping (see Figure 3 and lines 231–248, comparing GMPO$(e^{-0.4}, e^{0.4})$ with GMPO-seq-clip-$(e^{-0.4}, e^{0.4})$. This reduced variance leads to more stable policy optimization.
>
> **Q3: The major technical change from GRPO seems to be token-level clipping + geometric mean vs group arithmetic mean. The technical novelty is low.**
>
> **R3:** The core contribution of this paper is simple yet effective: replacing the arithmetic mean in GRPO with the geometric mean for aggregating importance-weighted rewards. Although technically straightforward, we provide substantial theoretical and empirical evidence showing that this small modification yields better performance and improved optimization stability.
>
> In the revised version, Appendix B further establishes the theoretical connection between GMPO and GRPO, proving that GMPO is a valid optimization objective. Appendix D additionally reports GMPO’s performance in agentic RL settings, demonstrating its effectiveness in broader application scenarios.
>
>
> **Q4: Although discussion is present, comparisons is missing against the method that controls training stability by carefully setting clipping ratio (e.g. DAPO)**
>
> **R4:** Thank you for the suggestion. We performed additional experiments on clipping ratios. We find that simply enlarging the clipping range in GRPO leads to training instability, causing performance fluctuations or even degradation. Applying a conservative “clip-higher’’ strategy (e.g., DAPO) does not yield noticeable improvements. Moreover, wider clipping thresholds further degrade GRPO performance. In contrast, broader thresholds can improve GMPO results, benefiting from the inherent stability of the geometric mean.
>
> | Model | Clipping thresholds  | AIME24 | AMC  | MATH500 | Minerva | Oly. | **Avg.** |
> |-----|---------------|--------|------|---------|---------|------|----------|
> | GRPO  | (0.8, 1.28) #clip-higher | 36.7  | 61.4 | 82.0    | 32.4   | 43.7 | 51.2     |
> | GRPO  | $(e^{-0.4}, e^{0.4})$ |  33.3  | 61.4  |  81.6   | 34.6  | 43.4 | 50.9     |
> | GRPO  | (0.8, 1.2)   | 40.0   | 59.0 | 83.4    | 32.4    | 41.3 | 51.2     |
> | GMPO | $(e^{-0.4}, e^{0.4})$     | 43.3   | 61.4 | 82.0    | 33.5    | 43.6 | 52.7     |

---

### Official Review · Reviewer_f72r · 2025-11-02

**Soundness:** 2
**Presentation:** 3
**Contribution:** 2
**Rating:** 4
**Confidence:** 4

**Summary:**

This paper addresses the critical challenge of training instability in GRPO. The authors identify the root cause of this instability as the GRPO objective's reliance on the arithmetic mean of token-level rewards, which is highly sensitive to outlier importance sampling ratios. To remedy this, the paper proposes Geometric-Mean Policy Optimization (GMPO), which replaces the arithmetic mean with the geometric mean. The central hypothesis is that the geometric mean, being inherently more robust to extreme values, will effectively dampen the influence of outlier tokens.

**Strengths:**

- The problem is clearly defined and motivated. The authors narrow their focus to a specific, plausible mechanism: the sensitivity of the arithmetic mean aggregator in the GRPO objective to these outlier IS ratios.

- The method is practical and easy to implement. The loss is a small rewrite of GRPO. The authors enhance this practicality by providing clear pseudo-code in Algorithm 1, which details the implementation in log-space for numerical stability.1 This transparency is crucial for reproducibility and allows the community to build upon the work easily.

- The paper substantiates its claims with extensive empirical evidence across a variety of benchmarks and model scales.

**Weaknesses:**

- The core novelty of the paper lies in the application of the geometric mean to the GRPO objective. While this application is new in this specific context, the underlying idea of using a robust statistical estimator to handle outliers is a foundational concept in statistics and data analysis. This weakness is compounded by a failure to justify the choice of the geometric mean over other standard robust estimators that designed to be less sensitive to outliers and could plausibly offer similar or even superior stabilization benefits.

- The theoretical analysis presented in Section 3 and Appendix A does not provide formal convergence guarantees, nor does it connect the method to more rigorous optimization frameworks that are commonly used to analyze the stability of policy gradient methods.

- A key design choice highlighted in the paper is the use of clipping range for the importance sampling ratio, which is substantially larger than the typical range used in GRPO. The authors justify this choice empirically, showing in their ablation study (Table 5) that this specific range yields the best performance among the options tested. However, the paper fails to provide a deeper explanation for why this is the case.

- The evaluation can be further enhanced. There are several alternatives to improve GRPO stability, such as GSPO, DAPO, or Dr.GRPO. Given how close GMPO is to sequence-ratio weighting and clip-widening, this is critical. Also, the evaluation is mostly math. It will be more convincing if the authors can demonstrate code tasks, tool-use, or larger open-ended reasoning. The improvement seems marginal.

**Questions:**

- Why does the geometric mean objective enable a wider clipping range to be effective without causing instability? Is there a principled way to determine the optimal clipping range in conjunction with GMPO, or is it simply another sensitive hyperparameter that requires careful tuning?

- Have the authors analyzed the computational and memory overhead of this operation compared to the simple summation in GRPO, particularly for very long reasoning chains (e.g., >2000 tokens)? Is there a point at which this overhead becomes a significant practical concern? A complete analysis would include profiling the computational time and memory usage of the GMPO loss function versus the GRPO loss function for varying sequence lengths to quantify any potential overhead.

---

> ### Author Response · Authors · 2025-11-21
> **Response to f72r (Part 1/2)**
>
> **Q1: The core novelty of the paper ... The choice of the geometric mean over other standard robust estimators.**
>
> **R1:** The main contribution of this work is to provide a new perspective on designing better and more stable RL algorithms for large language models (LLMs), i.e., **Mean matters in RL**: A more robust aggregation operator has the potential to improve stability and performance. While other standard robust estimators (aggregation operator) may also achieve strong results, we choose the geometric mean due to its simplicity, reduced sensitivity to extreme values compared to the original arithmetic mean used in GRPO, and favorable gradient properties.
>
> To further justify the choice of geometric mean over other standard robust estimators, we compare the performance and importance sampling ratio ranges of several classic reward aggregators, including a classic subset of power means (e.g., arithmetic, geometric, and harmonic means) as well as the interquartile mean. (**More details in Appendix C**) Their performances are as follows:
>
> | Reward Aggregator   | AIME24 | AMC  | MATH500 | Minerva | Oly. | **Avg.** |
> |---------------------|--------|------|---------|---------|------|----------|
> | Interquartile Mean  | 36.7   | 60.2 | 79.6    | 29.0    | 43.4 | 49.8     |
> | Arithmetic Mean     | 40.0   | 59.0 | 83.4    | 32.4    | 41.3 | 51.2     |
> | Harmonic Mean       | 36.7   | 56.6 | 83.4    | 36.0    | 45.9 | 51.7     |
> | Geometric Mean      | 43.3   | 61.4 | 82.0    | 33.5    | 43.6 | 52.7     |
>
> Among these choices, the geometric mean achieves the strongest overall performance and maintains stable optimization.
>
> **Q2: Formal convergence guarantees.**
>
> **R2:** We show that GMPO is an $O(\delta^{2})$ Lipschitz-stable perturbation of GRPO within a trust region, where $\delta$ is the maximum token-level ratio deviation. Consequently, GMPO maintains GRPO’s monotonic-improvement and convergence guarantees up to an $O(\delta^{2})$ error, making it a principled optimization objective. (**Detailed derivation in Appendix B**)
>
> **Q3: Provide a deeper explanation for large clipping range. Why does the geometric mean objective enable a wider clipping range to be effective without causing instability? Is there a principled way to determine the optimal clipping range in conjunction with GMPO**
>
> **R3:** Adjusting the clipping range represents a trade-off between exploration and stability. Specifically, increasing the clipping range for GRPO/PPO-like algorithms help maintain higher token entropy during training, which promotes greater policy exploration, making it easier for the policy model to converge to a better solution, as evidenced by DAPO[1]. However, if the clipping range is too large, policy updates may become unstable, as evidenced by the expanding importance sampling ratio (Figure 3 in the paper) or by violating the trust region constraint in TRPO[2]. Thus, adjusting the clipping range represents a trade-off between exploration and stability.
> *Since GMPO has lower variance in its importance sampling ratios and is more stable than GRPO, we can safely increase the clipping range to further enhance policy exploration and boost performance without sacrificing too much stability.*
>
> The method for determining the clipping range is shown in Figure 3 in the paper. Enlarging the clipping range leads to more instability during training, as indicated by a larger importance sampling ratio range. Therefore, we use a clipping range of ($e^{-0.4}$, $e^{0.4}$), which encourages policy exploration while maintaining stable training. Furthermore, the clipping ratio is not a sensitive hyperparameter that requires careful tuning according to Figure 3 and Table 4 in the paper. GMPO outperforms and is more stable than GRPO even without ratio clipping.
>
> [1] Yu, Qiying, et al. "Dapo: An open-source llm reinforcement learning system at scale." arXiv preprint arXiv:2503.14476 (2025).
>
> [2] Schulman, John, et al. "Trust region policy optimization." International conference on machine learning. PMLR, 2015.
>
>
> **Q4: There are several alternatives to improve GRPO stability.**
>
> **R4:** We additionally reproduced the performance of several alternatives including GSPO (under the same experimental setup as Dr. GRPO and GMPO in this paper). The final performance comparison is as the following table. It can be seen that GMPO significantly outperforms GSPO and Dr. GRPO by 1.3% and 1.3%.
>
> | Model   | AIME24 | AMC  | MATH500 | Minerva | Oly. | **Avg.** |
> |---------------------|--------|------|---------|---------|------|----------|
> | Qwen2.5-Math-7B  | 16.7   | 38.6 | 50.6    | 9.9    | 16.6 | 26.5     |
> | GRPO-7B  | 40.0   | 59.0 | 83.4    | 32.4    | 41.3 | 51.2     |
> | GSPO-7B  | 36.7   | 63.9 | 81.4    | 32.7    | 42.4 | 51.4     |
> | Dr. GRPO-7B (Oat-Zero-7B)     | 43.3   | 62.7 | 80.0    | 30.1    | 41.0 | 51.4     |
> | GMPO-7B (Ours)      | 43.3   | 61.4 | 82.0    | 33.5    | 43.6 | 52.7     |

---

> ### Author Response · Authors · 2025-11-21
> **Response to f72r (Part 2/2)**
>
> **Q5: It will be more convincing if the authors can demonstrate code tasks, tool-use, or larger open-ended reasoning.**
>
> **R5:** As suggested, we have applied GMPO in ALFWorld [3], which is a reasoning benchmark related to tool-use and embodied sequential reasoning. The results are as follows:
>
> | Agentic Model                               | Pick  | Look  | Clean | Heat  | Cool  | Pick2 | **ALL** |
> |--------------------------------------------|-------|-------|-------|-------|-------|-------|---------|
> | Qwen2.5-1.5B-Instruct             | 5.9   | 5.5   | 3.3   | 9.7   | 4.2   | 0.0   | 4.1     |
> | PPO-1.5B                     | 64.8  | 40.5  | 57.1  | 60.6  | 46.4  | 47.4  | 54.4    |
> | RLOO-1.5B                            | 88.3  | 52.8  | 71.0  | 62.8  | 66.4  | 56.9  | 69.7    |
> | GRPO-1.5B                        | 85.3  | 53.7  | 84.5  | 78.2  | 59.7  | 53.5  | 72.8    |
> | GiGPO-1.5B                          | 94.4  | 67.5  | 94.8  | 94.4  | 79.8  | 76.4  | 86.7|
> | GMPO-1.5B (Ours)                       | 93.1  | 78.6  | 81.0  | 88.2  | 82.1  | 89.5  | 85.9    |
>
> GMPO achieves a significant `13.1%` performance gain over GRPO on ALFWorld. Furthermore, GMPO demonstrates comparable performance with GiGPO (Group-in-group policy optimization for llm agent training, NeurIPS 2025), a method specifically designed for agentic RL tasks. (**More details in Appendix D**)
>
> [3] Shridhar, Mohit, et al. "ALFWorld: Aligning Text and Embodied Environments for Interactive Learning." International Conference on Learning Representations.
>
> **Q6: The computational and memory overhead of this operation compared to the simple summation in GRPO.**
>
> **R6:** Thank you for the suggestion. We profiled the overhead of GMPO relative to GRPO. On average, GMPO adds only approximately `6e-05 GB` of memory and `4e-05 seconds` of runtime per trajectory compared to GRPO (measured on a single H100 GPU). In practice, the additional cost is negligible, as GMPO only introduces one extra `log` and `exp` operation compared to GRPO.

---

### Meta-Review · Area_Chair_9Luw · 2026-01-05

**Summary:**

This paper proposes a Geometric Mean Policy Optimization (GMPO) method, a plug-and-play improvement to GRPO. GMPO replaces the arithmetic mean aggregation based on token-level importance-weighted rewards with a geometric mean, aiming to reduce sensitivity to outlier importance sampling ratios and stabilize updates. In the reported benchmarks, this paper claims that GMPO-7B achieves up to 4.1% improvement over GRPO on the Pass@1 metric, while also exhibiting superior stability metrics (e.g., ratio range and KL/entropy behavior). The authors also supplemented their response with agent/tool usage style evaluations in the ALFWorld environment, demonstrating that GMPO significantly improves upon GRPO in this environment.

**Reviewer Concerns:**

The initial reviewers focused on the following points: (i) limited novelty; (ii) whether the geometric mean was superior to other robust aggregators or stabilization methods; and (iii) insufficient depth and clarity in the theoretical exposition. To address these issues, the authors added direct comparisons with several classic aggregators (e.g., power-mean and interquartile-mean), expanded the clipping range analysis, and provided a lightweight performance analysis demonstrating minimal overhead on the testing hardware. They also clarified issues of novelty compared to contemporary work, which improved WSoW's score. VUr8 also explicitly stated that their concerns had been addressed and supported acceptance. One reviewer (f72r) maintained a score of 4, initially believing the paper lacked novelty and theoretical grounding, but did not strongly oppose acceptance.

**Reviewer Scores:**

I cannot reliably answer this counterfactual question without putting words in reviewers’ mouths. I will not impute score changes beyond what reviewers explicitly stated in the discussion. I instead provide a faithful synthesis of the discussion outcomes and remaining points of disagreement.

---

### Decision · Program_Chairs · 2026-01-26

Accept (Poster)